# Accelerating Monte Carlo Bayesian Prediction via Approximating Predictive Uncertainty over the Simplex

## Abstract

Estimating the predictive uncertainty of a Bayesian learning model is critical in various decision-making problems, e.g., reinforcement learning, detecting adversarial attack, self-driving car. As the model posterior is almost always intractable, most efforts were made on finding an accurate approximation the true posterior. Even though a decent estimation of the model posterior is obtained, another approximation is required to compute the predictive distribution over the desired output. A common accurate solution is to use Monte Carlo (MC) integration. However, it needs to maintain a large number of samples, evaluate the model repeatedly and average multiple model outputs. In many real-world cases, this is computationally prohibitive. In this work, assuming that the exact posterior or a decent approximation is obtained, we propose a generic framework to approximate the output probability distribution induced by model posterior with a parameterized model and in an amortized fashion. The aim is to approximate the true uncertainty of a specific Bayesian model, meanwhile alleviating the heavy workload of MC integration at testing time. The proposed method is universally applicable to Bayesian classification models that allow for posterior sampling. Theoretically, we show that the idea of amortization incurs no additional costs on approximation performance. Empirical results validate the strong practical performance of our approach.

## 1 Introduction

Bayesian inference is a principled method to estimate the uncertainty of probabilistic models. In most applications, especially in deep learning, the likelihood model and model prior are not conjugate hence marginalizing over model prior or posterior cannot be performed analytically, which hinders the practical applicability. For tractability, a simple point estimate such as maximum *a posteriori* (MAP) estimate could be used to approximate the full model posterior. The price paid is the loss of model uncertainty due to incomplete characterization of the model posterior. Approximate inference methods, such as Markov chain Monte Carlo and variational inference, enhance the approximate posterior by a better probability distribution while keeping inference tractability. However, even though a decent approximation of posterior can be obtained, computation of predictive distribution is usually intractable due to loss of conjugacy, and is of high cost if tractable.

To introduce the problem, we consider a Bayesian classification model trained on dataset $\mathcal{D} = \{(\mathbf{y}_n, \mathbf{x}_n)\}_{n=1}^N$, where $\mathbf{x}_n \in \mathcal{X}$ and $\mathbf{y}_n \in \mathcal{Y}$ are the $n$th input and output, respectively. Let the model posterior $p(\boldsymbol{\theta}|\mathcal{D})$ be approximated by MC estimate $\frac{1}{S}\sum_{s=1}^S \delta(\boldsymbol{\theta} - \boldsymbol{\theta}_s)$, and the predictive distribution (a categorical distribution parameterized by the predicted *class probabilities*) is thus approximated by

$$p(\mathbf{y}|\mathbf{x}, \mathcal{D}) = \int p(\mathbf{y}|\mathbf{x}, \boldsymbol{\theta}) p(\boldsymbol{\theta}|\mathcal{D}) \, \mathrm{d}\boldsymbol{\theta} \approx \frac{1}{S}\sum_{s=1}^S p(\mathbf{y}|\mathbf{x}, \boldsymbol{\theta}_s).$$

The predictive distribution can be accurately estimated as $S \to \infty$. However, to perform the computation, we need to maintain a large number of samples, repeatedly evaluate the model for $S$ times and finally average the model outputs. This problem is critical in many real-world cases. For example, assisted-driving car system requires an accurate measure of uncertainty to avoid making mistakes with a high confidence (Kendall & Gal, 2017; Sünderhauf et al., 2018). Due to the limited computational resources and storage in such system, it's hard to maintain a large number of samples and perform $S$ times evaluation of the Bayes model for the real-time image data.

In this work, aiming at boosting the prediction speed while maintaining a rich characterization of the prediction, we propose to approximate the *distribution of class probabilities* over the simplex induced by the model posterior $p(\boldsymbol{\theta}|\mathcal{D})$, in an amortized fashion. This naturally diverts the heavy-load MC integration process from testing period to approximation period. Different from the previous work in Bayesian knowledge distillation (Balan et al., 2015; Bulò et al., 2016) that only focuses on the output categorical distribution (a point on simplex), the induced distribution over simplex provides: 1) rich knowledge includes prediction confidence for identifying out-of-domain (OOD) data (see empirical examples in Fig. 2); 2) the possibility to use more expressive distributions as the approximate model.

We term the Bayes classifier as "Bayes teacher" and the approximate distribution as "student", due to the analogy with teacher-student learning. A Dirichlet distribution is used as the student due to its expressiveness, conjugacy to categorical distribution and its efficient reparameterization for training. We propose to explicitly disentangle the parameters of the student into a prediction model (PM) and concentration model (CM), which capture class probability and sharpness of Dirichlet respectively. The CM output can directly be used as a measure for detecting OOD data. We term our approximation method as One-Pass Uncertainty (OPU) as it simplifies real-world evaluation of Bayesian models by computing the predictive distribution with only one model evaluation. Note that, OPU allows choosing various types of student model (e.g., compressed neural network (Lin et al., 2017; Hubara et al., 2016; Han et al., 2015)) for further speedup on specific platforms with **no** extra design efforts.

As the amortized approximation of induced distributions is unexplored in the literature, we consider and compare several choices of probability distance measure: forward KL, earth-mover's distance (EMD) and maximum mean discrepancy (MMD). We theoretically analyze the performance gap incurred by the amortized approximation and show that, under MMD, besides model loss due to restriction of student distribution, the amortized approximation does not introduce additional loss.

Empirical evaluations show a significant speedup ($\sim 500\times$) of Bayes models. The results on Bayes NN show that OPU performs better in misclassification detection and OOD detection than state-of-the-art works in Bayesian knowledge distillation. It can also be observed that explicit disentangling of mean and concentration helps improve the performance. The comparisons of different probability measures validate the theoretical analysis. We also conduct empirical evaluations and comparisons on Bayes logistic regression and Gaussian process, to show OPU is universally applicable to all Bayesian classification models.

## 2 ONE-PASS UNCERTAINTY FRAMEWORK

### 2.1 INDUCED DISTRIBUTION OVER SIMPLEX

In this section we present our OPU framework for a generic Bayesian parametric classifier, e.g. Bayesian logistic regression (BLR) or Bayesian neural networks (NN). Let the Bayesian classifier be specified with categorical likelihood and a parametric function, i.e.,

$$p(\mathbf{y}|\mathbf{x}, \boldsymbol{\theta}) = \mathrm{Cat}(\mathbf{y}|\mathcal{T}(\mathbf{x}; \boldsymbol{\theta})) \tag{1}$$

and a prior distribution $p(\boldsymbol{\theta})$ be specified over the parameter space $\Theta$, where $\mathcal{T} : \mathcal{X} \times \Theta \to \mathcal{S}^{K-1}$ is a parametric function from input space to simplex, e.g., a neural network with softmax output layer, and $K$ is the number of classes. In this paper, we assume the posterior $p(\boldsymbol{\theta}|\mathcal{D})$ is obtained and focus on the computation of the predictive distribution. In what follows, let $p(\boldsymbol{\theta}|\mathcal{D})$ be approximate or exact (if available) model posterior, from which samples could be obtained.

From a dual perspective, with a fixed input $\mathbf{x}$, the mapping $\mathcal{T}_{\mathbf{x}} = \mathcal{T}(\mathbf{x}; \cdot)$ is also understood as a data-dependent mapping $\mathcal{T}_{\mathbf{x}} : \Theta \to \mathcal{S}^{K-1}$, which could be used to transform the posterior $p(\boldsymbol{\theta}|\mathcal{D})$ to a conditional distribution (on $\mathbf{x}$) over the simplex. That is, for each $\mathbf{x}$, we can define a random variable $\boldsymbol{\pi} = \mathcal{T}_{\mathbf{x}}(\boldsymbol{\theta})$ whose distribution is induced by $\boldsymbol{\theta} \sim p(\boldsymbol{\theta}|\mathcal{D})$ and $\mathcal{T}_{\mathbf{x}}$. This is effectively a well-defined push-forward measure $\mathcal{T}_{\mathbf{x}}\#p(\boldsymbol{\theta}|\mathcal{D})$ over the simplex (see Fig. 1). We term the conditional distribution of $\boldsymbol{\pi}|\mathbf{x}, \mathcal{D}$ as an "induced distribution" and define $p_{\mathbf{x}}(\boldsymbol{\pi}) = p(\boldsymbol{\pi}|\mathbf{x}, \mathcal{D})$ to keep the notation uncluttered.

The induced distribution isolates the dependence between input and output and simplifies the computation of predictive distribution due to the change-of-variable formula. Specifically, given a test point $\mathbf{x}^*$, the predictive distribution can be alternatively written as

$$p(\mathbf{y}|\mathbf{x}^*, \mathcal{D}) = \int_{\Theta} \mathrm{Cat}(\mathbf{y}|\mathcal{T}_{\mathbf{x}^*}(\boldsymbol{\theta}))p(\boldsymbol{\theta}|\mathcal{D}) \, \mathrm{d}\boldsymbol{\theta} = \int_{\mathcal{S}^{K-1}} \mathrm{Cat}(\mathbf{y}|\boldsymbol{\pi})p_{\mathbf{x}^*}(\boldsymbol{\pi}) \, \mathrm{d}\boldsymbol{\pi}. \tag{2}$$

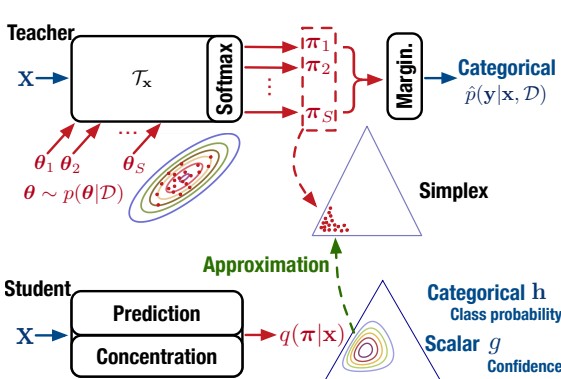

Figure 1: An intuitive example of the proposed framework. Only one input $\mathbf{x}$ is consider in this example. "Margin." indicates marginalization over $\boldsymbol{\pi}$.

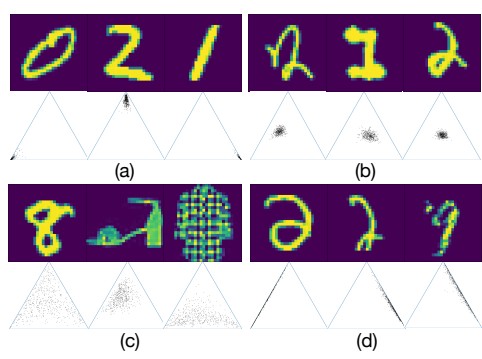

Figure 2: An empirical example of MC estimate of the induced distribution over simplex. The classifier (MCDP) is trained on real-world digits images of $\{0, 1, 2\}$ (corresponding to left, right, top corners of simplex). The 1st and 3rd rows indicate the input while the 2nd and 4th rows indicate MC estimate of $\boldsymbol{\pi}$ over a 2-simplex.

Our key insight is that $p_{\mathbf{x}}$ contains all information we need for prediction and uncertainty measurement. Hence $\boldsymbol{\pi}$ is sufficient in the sense that, given $\boldsymbol{\pi}$, $\mathbf{y}$ is independent of both $\boldsymbol{\theta}$ and $\mathbf{x}$. The isolation combines the complexities in both the likelihood and posterior into a single object $p_{\mathbf{x}}$, and keeps a simple dependence structure between $\mathbf{y}$ and $\boldsymbol{\pi}$. Also, the isolation renders the last probabilistic layer $\mathbf{y}|\boldsymbol{\pi}$ nuisance, which can thus be peeled off for prediction. To validate the idea, one can show that the predictive distribution is simply $p(\mathbf{y}|\mathbf{x}, \mathcal{D}) = \text{Cat}(\mathbf{y}|\mathbb{E}_{p_{\mathbf{x}}}\boldsymbol{\pi})$. The difference between probabilistic structures of original Bayesian model and the isolated version is showed in Fig. 3, Appendix.

Although the density function $p_{\mathbf{x}}$ is sometimes hard to compute, especially for complicated $\mathcal{T}_{\mathbf{x}}$ (like NN), its samples can be obtained via 1) sampling the posterior, i.e. $\boldsymbol{\theta}_s \sim p(\boldsymbol{\theta}|\mathcal{D})$; 2) "push-forward" the samples by $\mathcal{T}_{\mathbf{x}}$, i.e. $\boldsymbol{\pi}_s = \mathcal{T}_{\mathbf{x}}(\boldsymbol{\theta}_s)$. The behavior of $\boldsymbol{\pi}$ can be empirically observed via the *particles* $\{\boldsymbol{\pi}_s\}_{s=1}^S$. As shown in Fig. 2, $\boldsymbol{\pi}$ is able to distinguish different types of input $\mathbf{x}$ based on the behavior of its samples. In Fig. 2(a), the inputs are similar to the training data. The particles gather around a corner of the simplex, indicating a confident prediction. In Fig. 2(b), the inputs are digit images but are relatively hard to predict. Therefore, the particles gathers around the center, indicating the model is certain that the input is on decision boundary. In Fig. 2(c), the inputs are images outside the domain of training data, the particles spread over the simplex, which means the model has a high uncertainty about input, indicating out-of-domain (OOD). In Fig. 2(d), the particles spread along a line between two corners, indicating the model is confident that the result is not at the other corner.

However, using particles for inference is time-consuming as each particle requires one evaluation of the model. This motivates us to use a tractable conditional distribution $q(\boldsymbol{\pi}|\mathbf{x})$ to approximate $p_{\mathbf{x}}$.

## 2.2 AMORTIZED APPROXIMATION

The view of $p_{\mathbf{x}}$ enables flexible choices of $q(\boldsymbol{\pi}|\mathbf{x})$, as any distribution defined on $\mathbb{R}^K$ can be transformed to $\mathcal{S}^{K-1}$ via logistic transformation (Aitchison, 1982). However, modeling $q(\boldsymbol{\pi}|\mathbf{x})$ locally for every input is not practical, as the design efforts and number of parameters grows linearly with the number of data points. Therefore, we propose to approximate $p_{\mathbf{x}}$ in two aspects: 1) use a single family of distribution $q$; 2) to generalize to unseen examples, let the parameter of $q$, $\boldsymbol{\alpha}_{\mathbf{x}} = \boldsymbol{\alpha}(\mathbf{x}; \boldsymbol{\phi})$, be a function depending on $\mathbf{x}$ and parameterized by a set of global adaptive parameters $\boldsymbol{\phi}$, and thus $q_{\mathbf{x}} = q(\boldsymbol{\pi}|\mathbf{x}) = q(\boldsymbol{\pi}|\boldsymbol{\alpha}_{\mathbf{x}})$. The computational cost is amortized by casting the problem of learning a series of conditional distributions to a regression problem.

We term $p_{\mathbf{x}}$ as "teacher distribution" and $q_{\mathbf{x}}$ as "student". In our method, as seen in graphical representation in Fig. 3, Appendix, by proper approximation, the stochasticity and knowledge in node $\boldsymbol{\theta}$ of the teacher is transferred into node $\boldsymbol{\pi}$ of the student model, such that the full predictive uncertainty is maintained. In terms of computation, the approximation requires sampling only in the training stage. While in the testing stage, to obtain the predictive distribution, only one evaluation is required. Thus, we denote the framework as the One-Pass Uncertainty (OPU) model.

The above benefits do not introduce any concession on generalizability. Since OPU is based on approximating the distribution of the output class probabilities which is common for all classifiers,

the amortized approximation can be applied to any Bayesian classifier. Note that the approximation framework can be extended to non-parametric model like Gaussian process, where the computational cost of inference is high [1].

In this work, we choose the student model $q$ to be a Dirichlet distribution, $q(\boldsymbol{\pi}|\boldsymbol{\alpha_x}) = \mathrm{Dir}(\boldsymbol{\pi}|\boldsymbol{\alpha_x})$, where $\boldsymbol{\alpha_x}$ is a function mapping input $\mathbf{x}$ to a Dirichlet parameter. The reasons for choosing Dirichlet is the tractability: the Dirichlet is the conjugate prior to the Categorical, and thus enables tractable integration of (2) given the parameters. To better disentangle the uncertainty measures, we use the design $\boldsymbol{\alpha_x} = \mathbf{h_x} \cdot \mathrm{e}^{g_x}$, where $\mathbf{h_x} = \mathbf{h}(\mathbf{x}; \boldsymbol{\phi}_1)$ and $g_\mathbf{x} = g(\mathbf{x}; \boldsymbol{\phi}_2)$ are two neural networks, and the vector output $\mathbf{h}$ sums to 1. Vector output $\mathbf{h}$ determines the mean of the Dirichlet (i.e., the predicted class probabilities), and $g$ determines the concentration of the Dirichlet (i.e., the prediction confidence). To see this, the posterior of the class labels is the Dirichlet mean, $p(y_\ell = 1|\mathbf{x}, \boldsymbol{\phi}) = \int \mathrm{Cat}(y_\ell = 1|\boldsymbol{\pi})\mathrm{Dir}(\boldsymbol{\pi}|\boldsymbol{\alpha_x})\,\mathrm{d}\boldsymbol{\pi} = \frac{\alpha_{\mathbf{x},\ell}}{\sum_c \alpha_{\mathbf{x},c}} = h_{\mathbf{x},\ell}$ where $y_\ell$ and $\alpha_{\mathbf{x},\ell}$ are the $\ell$-th coordinate of $\mathbf{y}$ and $\boldsymbol{\alpha_x}$ respectively. Therefore, we call $\mathbf{h_x}$ as the "prediction model" (PM). Similarly, the precision parameter $\alpha_0$ (determines sharpness) of the Dirichlet solely depends on $g_\mathbf{x}$, $\alpha_0 = \sum_c \alpha_{\mathbf{x},c} = \sum_c h_{\mathbf{x},c}\mathrm{e}^{g_x} = \mathrm{e}^{g_x}$. Therefore, we term $g_\mathbf{x}$ as the "concentration model" (CM). Based on this property, whether the Dirichlet is flat or not, can be fully characterized by CM. It can be expected that, when approximating particles in Fig. 2(a) and (b), the output value of CM is high, as the samples are concentrated, which means high confidence. CM outputs a low value for particles in Fig. 2(c), yielding a flat distribution and low confidence.

## 2.3 Learning

With a probability distance or divergence, we define the approximation loss $\mathcal{L}(\mathbf{x}, \boldsymbol{\alpha}) = \rho(p_\mathbf{x}, q_\mathbf{x})$ The student model is trained to minimize the aggregated objective

$$\min_{\boldsymbol{\alpha} \in \mathcal{F}} \quad \mathbb{E}_{p(\mathbf{x})}\mathcal{L}(\mathbf{x}, \boldsymbol{\alpha}) \tag{3}$$

where $\mathcal{F}$ is some hypothesis space and $p(\mathbf{x})$ is some distribution over $\mathcal{X}$. In practice we take $p(\mathbf{x}) = p_{\mathcal{D}'}(\mathbf{x})$ with $\mathcal{D}' = \{\mathbf{x}_m\}_{m=1}^M$ a dataset containing features only. As the amortized approximation of induced distributions in Bayesian classifiers is unexplored in the literature, we consider and compare several choices of $\rho$ including KL divergence, earth mover's distance (EMD) and maximum mean discrepancy (MMD). The corresponding derivation of the training objectives and training algorithms are in the appendix.

**Forward KL divergence** Minimizing aggregated reverse KL divergence is not tractable in our scenario as the density function $p_\mathbf{x}$ is not available in general. This difficulty is avoided by using forward KL, in which the intractable density function is only involved in the irrelevant entropy term. It is equivalent to using cross-entropy as a local loss, i.e. $\mathcal{L}(\mathbf{x}, \boldsymbol{\alpha}) = -\mathbb{E}_{p_x} \ln q_\mathbf{x}$. By plugging in a particle estimation $\hat{p}_\mathbf{x} = \frac{1}{S}\sum_{s=1}^S \delta(\boldsymbol{\pi} - \mathcal{T}_\mathbf{x}(\boldsymbol{\theta}_s))$, the training objective becomes $\min_{\boldsymbol{\alpha}} -\mathbb{E}_{p_{\mathcal{D}'}(\mathbf{x})} \frac{1}{S}\sum_s \ln q_\mathbf{x}(\mathcal{T}_\mathbf{x}(\boldsymbol{\theta}_s))$, which is equivalent to an "amortized" MLE problem with particles providing the estimation of sufficient statistics of $q_\mathbf{x}$. Due to the zero-avoiding nature, forward KL tends to over-estimate the support of $p_\mathbf{x}$. This leads to under-confidence approximation ("flat" approximate distribution) and hence might deteriorate the quality of uncertainty measurements. This is expected to be more serious when $p_\mathbf{x}$ is multi-modal.

**EMD** It is known that EMD provides much weaker topology than other probability distance measures Peyré et al. (2019). In the application where data is supported on strictly lower-dimensional manifolds, EMD provides more stable gradient than KL divergence (Arjovsky et al., 2017; Tolstikhin et al., 2017). An example of particles on low-dimensional manifold is shown in Fig. 2(d).

Specifically, the KR dual representation of EMD (Villani, 2008) in our problem is given by $W_1(p_\mathbf{x}, q_\mathbf{x}) = \sup_{\|\psi\|_\mathrm{L} \leq 1} \mathbb{E}_{\boldsymbol{\theta} \sim p(\boldsymbol{\theta}|\mathcal{D})}[\psi(\mathcal{T}_\mathbf{x}(\boldsymbol{\theta}))] - \mathbb{E}_{\boldsymbol{\pi} \sim q_\mathbf{x}}[\psi(\boldsymbol{\pi})]$, where $\|\cdot\|_\mathrm{L}$ denotes the Lipschitz semi-norm and $\psi$ is known as the *critic* (or the discriminator (Arjovsky et al., 2017)). As the induced distribution is conditioned on $\mathbf{x}$, a local critic $\psi_\mathbf{x}$ should be defined for each $\mathbf{x}$. Following Arjovsky et al. (2017), intractable supremum is solved by parameterizing $\psi_\mathbf{x} = \psi(\cdot; w_\mathbf{x})$. To avoid training $|\mathcal{D}'|$ local critics, we propose to let $w$ be the global weight, and let $\psi$ depends on $\mathbf{x}$, i.e., $\psi_\mathbf{x} = \psi(\boldsymbol{\pi}; w, \mathbf{x})$ (see Fig. 4, Appendix). The final aggregated training objective becomes,

$$\min_{\boldsymbol{\alpha}} \max_w \quad \mathbb{E}_{p_{\mathcal{D}'}(\mathbf{x})}\left[\mathbb{E}_{p(\boldsymbol{\theta}|\mathcal{D})}[\psi(\mathcal{T}_\mathbf{x}(\boldsymbol{\theta}); w, \mathbf{x})] - \mathbb{E}_{q_\mathbf{x}}[\psi(\boldsymbol{\pi}; w, \mathbf{x})]\right] + \lambda\mathcal{R}(w), \tag{4}$$

---

[1]The details of extracting samples from various Bayes parametric models and non-parametric model (GP classifier) are summarized in the appendix.

where $w$ is the introduced global parameter for $\psi$, $\mathcal{R}(w)$ is the imposed gradient penalty (Gulrajani et al., 2017) over $w$ to enforce the Lipschitz constraint. Solving the minimax problem requires the supremum to be attained for each $\mathbf{x}$ under the Lipschitz constraint. Specifically, in every optimization step, $\psi$ is trained to generate a critic for each $\mathbf{x}$ that matches the exact EMD. Practically, this needs a high-capacity critic and the required capacity increases with the number of classes $K$.

**MMD** Let $\mathcal{H}_k$ be a reproducing kernel Hilbert space (RKHS) defined by a positive-definite kernel $k$, the MMD between $p_{\mathbf{x}}$ and $q_{\mathbf{x}}$ can be written as

$$\mathrm{MMD}_k(p, q) = \sup_{\psi \in \mathcal{H}_k, \|\psi\|_{\mathcal{H}} \leq 1} \mathbb{E}_{p(\boldsymbol{\theta}|\mathcal{D})}[\psi(\mathcal{T}_{\mathbf{x}}(\boldsymbol{\theta}))] - \mathbb{E}_{q_{\mathbf{x}}}[\psi(\boldsymbol{\pi})] \tag{5}$$

Compared with EMD, the advantage of MMD is that there is no need to train an NN as the critic that maximizes Eq. 5. With kernel trick, MMD can be readily estimated in closed-form with its empirical version under finite sample (Sec. C).

Compared with KL divergence, MMD is a valid statistical metric. Due to the symmetry property, the approximation is expected to be neither mean-seeking nor mode-seeking. Therefore, MMD is not expected to have an under-confidence issue.

**Reparameterization** Note that optimization under both EMD and MMD requires gradient of the expectation of critic via sampling from $q$, which contains parameters. To obtain efficient gradient estimator and reduce variance, we use the reparameterization trick (specifically, implicit reparameterization trick). For details, see Sec. C in Appendix.

## 2.4 Amortization Gap

To better understand nature of the proposed approximation, we consider the "unamortized" version of the approximation as an intermediate stage, which involves fitting separate approximations to each $p_{\mathbf{x}}$. To demonstrate the idea, we leverage to MMD due to its nice property. For fixed $p_{\mathbf{x}}$, the optimal point-wise approximation within family $\mathcal{Q}$ is defined as

$$\bar{q}_{\mathbf{x}}^* = \arg\min_{q \in \mathcal{Q}} \mathrm{MMD}_k(q, p_{\mathbf{x}}). \tag{6}$$

Then we have the following lemma:

**Lemma 1.** *Let $\mathcal{P}(\mathcal{S}^{K-1})$ be the space of probability measures over the simplex, equipped with MMD metric defined by a universal kernel. If $\mathcal{T}$ satisfies Assumption 1, then the map $\mathbf{x} \mapsto p_{\mathbf{x}}$ is continuous. Further, if $\mathcal{Q} \subseteq \mathcal{P}(\mathcal{S}^{K-1})$ is a closed convex model space, the projection $\bar{q}_{\mathbf{x}}^*$ is unique and the map $\mathbf{x} \mapsto \bar{q}_{\mathbf{x}}^*$ is also continuous.*

Further, if we assume the model space is parameterized and identifiable in MMD, i.e. $\mathcal{Q} = \{q(\boldsymbol{\pi}|\boldsymbol{\alpha}) : \boldsymbol{\alpha} \in \mathcal{A}\}$ and $\mathrm{MMD}_k(q(\boldsymbol{\pi}|\boldsymbol{\alpha}), q(\boldsymbol{\pi}|\boldsymbol{\alpha}')) = 0$ if and only if $\|\boldsymbol{\alpha} - \boldsymbol{\alpha}'\| = 0$, we may obtain continuity in parameter space. The continuity of optimal parameters implies there exists a continuous function $\boldsymbol{\alpha}^* : \mathbf{x} \in \mathcal{X} \mapsto \boldsymbol{\alpha}^*(\mathbf{x}) \in \mathcal{A}$, which serves as the essential target of our amortized goal.

To analyze how amortization affects the approximation, we define the local amortization gap as

$$\Delta(\mathbf{x}) = \mathrm{MMD}_k(q_{\mathbf{x}}, p_{\mathbf{x}}) - \mathrm{MMD}_k(\bar{q}_{\mathbf{x}}^*, p_{\mathbf{x}}). \tag{7}$$

Then it holds that

$$0 \leq \Delta(\mathbf{x}) \leq \mathrm{MMD}_k(q_{\mathbf{x}}, \bar{q}_{\mathbf{x}}^*), \tag{8}$$

where the lower bound is because $\bar{q}_{\mathbf{x}}^*$ is the projection and the upper bound is due to triangle inequality. Then our goal, minimizing aggregated MMD loss $\mathbb{E}_{p(\mathbf{x})}\mathrm{MMD}_k(q_{\mathbf{x}}, p_{\mathbf{x}})$, is essentially equivalent to minimizing aggregated amortization gap $\mathbb{E}_{p(\mathbf{x})}\Delta(\mathbf{x})$, up to an irrelevant additive constant.

One can see that $\boldsymbol{\alpha}^*$ is the global minimizer of aggregated amortization gap and hence the global minimizer of the aggregated MMD loss. Intuitively, if $\mathcal{F}$ is of enough capacity, then the infimum (over $\mathcal{F}$) of aggregated amortization gap could be 0, i.e. we can push the amortization gap arbitrarily small. If the optimal solution is covered by hypothesis space, no additional cost is introduced by amortizing the approximation. In other words, model loss due to using restrictive $\mathcal{Q}$ will dominate. Further, if the global optimum is reached, OPU approximation exactly matches the point-wise minimizer, i.e. $\mathrm{MMD}_k(q_{\mathbf{x}}, q_{\mathbf{x}}^*) = 0$, due to uniqueness of projection.

## 3   RELATED WORK

In this section, related works are reviewed and compared with OPU in terms of methodology. In CompactApprox (Snelson & Ghahramani, 2005), a parametric model composed of a small subset of "best samples" selected from the original MC samples is used to approximate the full predictive Categorical distribution. Extending the approximation to Bayes NN, BDK (Balan et al., 2015) proposes to use NN to approximate the predictive Categorical distribution of a Bayes NN trained by stochastic gradient Langevin dynamics (SGLD). Specifically, the teacher network generates samples via SGLD and KL between the two distributions is minimized in an online fashion. However, the disadvantage is that data uncertainty, model uncertainty and distributional uncertainty are all entangled in the class probabilities because categorical distributions are of limited expressiveness.

Different from these previous works that only approximate the class probabilities, OPU approximates the induced *distribution* of class probabilities, which contains richer information including both class probabilities and prediction confidence (e.g., the three types of uncertainty observable via the samples in Fig. 2.) We choose a Dirichlet distribution as the student model, and explicitly disentangle the mean and concentration to fully capture the thee types of uncertainty. We also explore other probability distance measures (EMD and MMD), showing that KL yields degenerate prediction performance.

Using a Dirichlet to estimate uncertainty has also been explored by Deep Prior Network (DPN) (Malinin & Gales, 2018), where a parameterized Dirichlet is used in a Bayesian model to characterize the "distributional uncertainty", i.e., to tell if the data is in the training domain or not. However, DPN adds a stochastic layer in the Bayes model, rather than approximating a well-trained Bayes teacher. Due to the intractable inference, DPN uses an MAP estimate of the model posterior which incurs a *loss of uncertainty*. To compensate for the lost characterization of uncertainty, DPN uses a hand-crafted training goal that explicitly requires OOD examples (which are typically unavailable in real-world applications). In contrast to DPN, our OPU is able to: 1) extract predictive uncertainty from any Bayesian classification model according to the practical requirements; 2) choose various types for student model (e.g., quantized neural network) to enable fast prediction; 3) use only in-domain data in training to get a good uncertainty measure (see Sec. 4). Note that **none** of these properties can be achieved by DPN.

## 4   EXPERIMENTS

### 4.1   EXPERIMENTAL SETUP

**Models and Tasks**   In this section, we present the experimental results on using Bayesian NN (BNN).[2] For each model, we choose a few Bayesian methods as teachers, and approximate them with our OPU. We also compare with state-of-the-art approximations that are proposed for the specific types of methods. For BNN, we use MCDP (Gal & Ghahramani, 2016) and SGLD (Welling & Teh, 2011) as the teacher, and BDK and DPN for comparisons. The methods for obtaining posterior samples from the Bayes teachers are in Appendix B. For each type of model, in-domain misclassification (MisC) detection, out-of-domain (OOD) input detection, prediction performance and prediction time are presented.

**Baselines and Uncertainty Measures**   For the uncertainty measures of teachers, we take MCDP and KL as a example. For each testing data $\mathbf{x}$, a Dirichlet is fit on particles under KL, to get an optimal $q_{\mathbf{x}}^*$, whose differential entropy (D) is an uncertainty measure. For the uncertainty in prediction, to avoid the model loss, entropy (E) and maximum probability (P) of the averaged particle are directly adopted as uncertainty measures. This gives MCDP-KL model as a baseline. The other baselines MCDP-EMD and MCDP-MMD are also obtained similarly. Note that they share the same E and P as the sample mean is the same. The D of $q_{\mathbf{x}}^*$ is expected to be the best uncertainty measure that OPU can approach theoretically (when the amortization loss is zero, see Sec. 2.4). For fairness, the same set of posterior particles is used for obtaining D. The baselines for SGLD are obtained in a similar way. Students use categorical entropy (for BDK) or D (for DPN) and P of the output distributions. For OPU, we consider E and P of the prediction model, and the scalar output of the concentration model (C) as the measures.

**Data and Evaluation Metrics**   For a fair comparison, we let $\mathcal{D}' = \mathcal{D}$ in the approximation of OPU. The in-domain dataset is split to training data and testing data, i.e., $\mathcal{D}^{\text{in}} = \mathcal{D}' = \{\mathcal{D}^{\text{tr}}, \mathcal{D}^{\text{te}}\}$, which is used for training models, evaluating prediction and MisC detection. The OOD dataset $\mathcal{D}^{\text{ood}}$ and $\mathcal{D}^{\text{te}}$ are used for OOD detection. To assess the performance, we use accuracy, time, Area under the ROC

---

[2]The setup and results on BLR and GP are available in the appendix.

Table 1: Results on BNN - MNIST. The doubt quote means "same as above".

| Model | MisC detection | | Omniglot | | SEMEION | | Acc. | Test |
|---|---|---|---|---|---|---|---|---|
| | AUROC | AUPR | AUROC | AUPR | AUROC | AUPR | (%) | time(s) |
| MCDP-KL | 97.3 (E) | 43.0 (E) | 99.4 (D) | 99.7 (D) | 86.8(E) | 53.8 (P) | 97.9 | 210.6 |
| MCDP-EMD | 97.3 (E) | 43.0 (E) | 99.6 (D) | 99.9 (D) | 86.8(E) | 53.8 (P) | 97.9 | " |
| MCDP-MMD | 97.3 (E) | 43.0 (E) | 99.7 (D) | 99.9 (D) | 90.1 (D) | 71.2 (D) | 97.9 | " |
| OPU-MCDP-KL | 94.2 (E) | 37.7 (E) | **100** (C) | 77.0 (C) | 91.4 (C) | 67.3 (C) | 96.2 | **0.443** |
| OPU-MCDP-EMD | 95.3 (P) | **43.7** (P) | **100** (C) | **100** (C) | 93.3 (C) | 82.5 (C) | 96.1 | " |
| OPU-MCDP-MMD | **97.2** (P) | 40.9 (P) | **100** (C) | **100** (C) | **99.8** (C) | **98.6** (C) | **97.9** | " |
| SGLD-KL | 97.9 (P) | 46.2 (E) | 99.2 (E) | 99.6 (E) | 89.3 (E) | 46.8 (E) | 98.4 | 233.5 |
| SGLD-EMD | 97.9 (E) | 46.2 (E) | 99.4 (D) | 99.7 (D) | 89.9 (D) | 47.1 (D) | 98.4 | " |
| SGLD-MMD | 97.9 (E) | 46.2 (E) | 99.2 (E) | 99.6 (E) | 89.3 (E) | 46.8 (E) | 98.4 | " |
| OPU-SGLD-KL | 94.2 (E) | **46.7** (E) | **100** (C) | **100** (C) | **99.5** (C) | **98.4** (C) | **98.2** | 0.443 |
| OPU-SGLD-EMD | 93.7 (P) | 44.4 (E) | **100** (C) | **100** (C) | 98.9 (C) | 96.2 (C) | 98.0 | " |
| OPU-SGLD-MMD | **97.2** (P) | 44.6 (E) | **100** (C) | **100** (C) | 99.1 (C) | 98.0 (C) | 98.1 | " |
| BDK-SGLD | 85.9 (E) | 46.6 (E) | 46.1 (E) | 41.7 (E) | 35.3 (P) | 46.5 (P) | 92.1 | **0.441** |
| BDK-MCDP | 86.9 (E) | 41.1 (E) | 47.5 (P) | 44.1 (P) | 43.3 (P) | 47.2 (P) | 92.4 | " |
| BDK-DIR-SGLD | 89.9 (E) | 40.0 (E) | 95.4 (E) | 96.4 (E) | 74.7 (E) | 38.3 (E) | 94.1 | " |
| DPN | 99.0 (E) | 43.6 (E) | 100 (E) | 100 (E) | 99.7 (E) | 98.6 (E) | 99.4 | — |

Table 2: Results on BNN - Balanced EMNIST.

| Model | MisC detection | | Omniglot | | Acc. |
|---|---|---|---|---|---|
| | AUROC | AUPR | AUROC | AUPR | (%) |
| MCDP-KL | 89.7 (P) | 46.8 (P) | 99.7 (E) | 99.7 (E) | 88.8 |
| MCDP-MMD | 89.7 (P) | 46.8 (P) | 99.9 (D) | 99.9 (D) | 88.8 |
| OPU-MCDP-KL | 84.8 (P) | 40.6 (P) | 96.2 (E) /67.5 (C) | 96.5 (E) /63.7 (C) | 87.9 |
| OPU-MCDP-MMD | **89.8** (P) | **49.6** (P) | **100.0** (C) | **100** (C) | **88.4** |

(AUROC) and PR (AUPR), following the baseline in (Hendrycks & Gimpel, 2017) (The detailed calculation process is shown appendix D.3). Time is evaluated on the whole $\mathcal{D}^{\text{te}}$. To save space, we only present the best performing uncertainty measure (E, P or C) for each task and method. (Full results on undistilled MCDP/SGLD are shown in appendix D.3.) We use the MXNet implementation of BDK and GPflow implementation of GP, and the remaining models are implemented in Pytorch. All experiments run on a desktop with an i7-8700 CPU and an RTX-2080 Ti GPU. The experiments for Bayesian logistic regression follows the same setup as Gaussian process.

## 4.2 BAYESIAN NEURAL NETWORK

The experiments for Bayesian neural network use MNIST and balanced EMNIST datasets as $\mathcal{D}^{\text{in}}$, and use Omniglot and SEMEION dataset as $\mathcal{D}^{\text{ood}}$, as in Malinin & Gales (2018). Given MCDP and SGLD as teacher models, the baselines are obtained as illustrated in Sec. 4.1. The rest models are: OPU approximating MCDP (OPU-MCDP), OPU-SGLD, BDK-SGLD, BDK-Dir-SGLD and DPN. For BDK-Dir-SGLD, we replace the Categorical distribution in BDK by a Dirichlet without disentangling the mean and concentration, then train it with the same MC ensemble as OPU-SGLD. This is to show the benefits of explicit disentanglement.

The NN architecture used by these models is an MLP with size 784-400-400-10, ReLU activations, and softmax outputs, following Balan et al. (2015). For CM, we use an MLP with size 784-400-400-1. MCDP is trained by SGD with hyper-parameters: dropout-rate of 0.5, learning rate $5 \times 10^{-4}$, mini-batch size of 256, number of iterations $10^3$. The parameters of critic are shown in Appendix. For MMD, we use a summation of RBF kernel and polynomial kernel. OPU-MCDP is trained by Adam with hyper-parameters: number of iterations 100, learning rate for student $10^{-3}$. The training of SGLD and BDK follows Balan et al. (2015). Then OPU-SGLD is trained with the same hyperparameters as OPU-MCDP. Results of DPN are from Malinin & Gales (2018). The results are presented in Table 1.

**Computation time.** OPU offers a $\sim$500x speedup compared to the original MCDP/SGLD, as OPU only evaluate the model twice (PM and CM in the student network) while MCDP/SGLD evaluates for $S$ times. This confirms our idea of accelerating Bayesian prediction by *diverting the sampling process from the test period to the approximation period*. Note that the time cost of MCDP/SGLD increases with more posterior samples involved. BDK is slightly faster than OPU because it runs one network while OPU runs both PM and CM. Experimental results on original MCDP/SGLD with fewer samples are given in appendix D.3 to show performance degradation with fewer samples.

**OPU vs BDK.** In some tasks especially OOD detection, the measure of concentration outperforms the baseline. This is because the explicit disentanglement of mean and concentration helps "targeted" knowledge distillation, as shown in Sec. 2.2. By comparing OPU-SGLD-KL and BDK (trained by forward KL), we observe that OPU-SGLD-KL is significantly better in OOD detection tasks and AUROC in MisC detection. BDK shows a slight advantage in AUPR in MisC detection task. This is because the knowledge distillation only happens between two categorical variables in BDK, which only helps capturing prediction information. In contrast, OPU framework first extracts all information in a BNN with the induced distribution, then transfers the knowledge to a more expressive distribution with a small loss guaranteed (Sec. 2.4). Adding a Dirichlet distribution to BDK (BDK-DIR-SGLD) helps improving the performance in OOD detection. However, on SEMEION, which is expected to be harder as it is more similar to MNIST, there is a large performance difference from OPU. This further validates the necessity of explicit disentanglement of mean and concentration.

**OPU vs DPN.** Our OPU model (without OOD data in training) has comparable performance to DPN (which uses a hand-crafted goal and OOD data in training). Another reason that DPN performs slightly better is that DPN uses VGG-6 (4 Convolutional layer and 1 FC layer), which is a much stronger model than the 2-layer MLP model that other models use.

**KL vs EMD vs MMD.** With MCDP, MCDP-MMD gives the best performance of differential entropy baseline (D). This is because the samples of MCDP are relatively spread out over the simplex and might be multi-modal. The probability distance is fully captured by MMD under such case. EMD is expected to perform well as there are a lot of samples $\pi_s$ residing on a low-dimensional manifold. However, the performance seems to be degenerated due to the limited capacity of the hyper-network and the difficulty to train the minimax problem. KL presents the worst performance as expected because it is likely to be under confident with samples of MCDP. With SGLD, the performance of KL is the best except for AUROC of MisC detection. This is because the samples $\pi_s$ of SGLD over the simplex are much denser and are typically unimodal.

**KL vs MMD (EMNIST).** The experiments are conducted on EMNIST whose number of classes is large. We choose Omniglot as the OOD detection dataset as it is also contains handwritten characters, which are harder than SEMEION for a model trained on EMNIST. A CNN with structure similar with LeNet is used for MCDP and OPU (20 and 50 output channels in two convolutional layers). The baseline approach achieves a classification accuracy of 88.8%. The samples of MCDP Bayes NN are expected to be even more dispersive and multi-modal than MCDP trained with MNIST. OPU trained with EMD failed to converge, possibly because the capacity of the hypernetwork was not enough. Therefore, we do not recommend to use EMD for amortized approximation of predictive uncertainty, unless a more scalable estimator of EMD can be provided. As shown in Table 2, the performance gap between OPU-MCDP-KL and the baseline is larger because the multi-modality is severe – the entropy of sample mean is a better measure for OOD detection than the concentration. This further validates that OPU trained by KL suffers from an under-confidence issue. Specifically, KL forces OPU to cover the support of all samples, making the student distribution more dispersive. This inaccurate estimate of concentration affects the estimation of prediction results (mean) in turn, thus the accuracy is lower and MisC detection performance is degenerated. By contrast, MMD consistently provides an approximation that has similar performance with the teacher, that echoes the analysis in Sec. 2.4.

**Amortization Gap.** The details of experiments on the approximation error/gap can be found in appendix D.3. It is shown that the amortization gap of using OPU is small, indicating the function space is enough to cover the essential target. It also shows the approximation error is mainly from the model error of using Dirichlet distribution, which is small as well, see appendix D.3 for details.

## 5 DISCUSSION

The idea of "transferring" the randomness from model posterior to a simple-structure distribution at the output can be generalized to other problems where a real-time evaluation of uncertainty is critical, e.g., object segmentation. This allows interesting designs of structured output distributions.

The choice of student distribution for OPU is flexible. Although we verify that Dirichlet is suitable for the tested classification tasks, there are still chances that Dirichlet incurs great model error. Note that, in our framework, using a Dirichlet for the student is modeling choice, similar to assuming Gaussian posteriors for variational approximations. The OPU framework is general and any student distribution can be adopted, e.g., generalized Dirichlet, mixture of Dirichlets. To do this requires: 1) a suitable parametrization of the model that can capture uncertainty; 2) deriving/computing the approximation loss (KL, MMD, EMD); 3) reparameterization trick of expectations for efficient gradient estimation. The algorithms of KL, EMD and MMD as well as the analysis still apply.

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

## A    APPENDIX: PROOF

**Assumption 1.** *Let $\mathcal{T} : \mathcal{X} \times \Theta \to \mathcal{Y}$ be a map between finite dimensional vector spaces. We say $\mathcal{T}$ satisfies Assumption 1 for distribution $p$ if $\mathcal{T}(\cdot; \boldsymbol{\theta})$ is Lipschitz and the Lipschitz constant $L_{\boldsymbol{\theta}}$ satisfies $\mathbb{E}_{\boldsymbol{\theta} \sim p} L_{\boldsymbol{\theta}} < +\infty$.*

*Proof.* We simply show both maps are Lipschitz continuous with MMD metric on $\mathcal{P}(\mathcal{S}^{K-1})$. Let $\mathbf{x}, \mathbf{x}' \in \mathcal{X}$ and $\psi \in \mathcal{H}_k$ such that $\|\psi\|_{\mathcal{H}_k} \leq 1$. For $\mathbf{x} \mapsto p_{\mathbf{x}}$, the "push-forward" definition of $p_{\mathbf{x}}$ leads to

$$
\begin{aligned}
\mathrm{MMD}_k(p_{\mathbf{x}}, p_{\mathbf{x}'}) &= \sup_{\|\psi\| \leq 1} \mathbb{E}_{p(\boldsymbol{\theta}|\mathcal{D})} \psi(\mathcal{T}_{\mathbf{x}}(\boldsymbol{\theta})) - \mathbb{E}_{p(\boldsymbol{\theta}|\mathcal{D})} \psi(\mathcal{T}_{\mathbf{x}'}(\boldsymbol{\theta})) \\
&\leq \sup_{\|\psi\| \leq 1} \mathbb{E}_{p(\boldsymbol{\theta}|\mathcal{D})} |\psi(\mathcal{T}_{\mathbf{x}}(\boldsymbol{\theta})) - \psi(\mathcal{T}_{\mathbf{x}'}(\boldsymbol{\theta}))| \\
&\leq \sup_{\|\psi\| \leq 1} \|\psi\| \mathbb{E}_{p(\boldsymbol{\theta}|\mathcal{D})} d_k(\mathcal{T}_{\mathbf{x}}(\boldsymbol{\theta}), \mathcal{T}_{\mathbf{x}'}(\boldsymbol{\theta})) \\
&\leq C \mathbb{E}_{p(\boldsymbol{\theta}|\mathcal{D})} \|\mathcal{T}_{\mathbf{x}}(\boldsymbol{\theta}) - \mathcal{T}_{\mathbf{x}'}(\boldsymbol{\theta})\| \\
&\leq C \mathbb{E}_{p(\boldsymbol{\theta}|\mathcal{D})} L_{\boldsymbol{\theta}} \|\mathbf{x} - \mathbf{x}'\|,
\end{aligned}
$$

where $d_k = $ is a kernel-based distance. The constant $C$ emits when bounding $d_k$ with Euclidean norm and the existence of such a constant is due to topology equivalence in finite-dimensional space. For $\mathbf{x} \to \bar{q}_{\mathbf{x}}^*$, we notice that $\bar{q}_{\mathbf{x}}^*$ is the projection of $p_{\mathbf{x}}$ onto $\mathcal{Q}$ (effectively the projection of kernel mean embeddings in $\mathcal{H}_k$). By projection theorem in Hilbert space, the projection map $p_{\mathbf{x}} \mapsto \bar{q}_{\mathbf{x}}^*$ is non-expansive, i.e.

$$
\mathrm{MMD}_k(\bar{q}_{\mathbf{x}}^*, \bar{q}_{\mathbf{x}'}^*) \leq \mathrm{MMD}_k(p_{\mathbf{x}}, p_{\mathbf{x}'}),
$$

which leads to Lipschitz property of $\mathbf{x} \mapsto \bar{q}_{\mathbf{x}}^*$. $\qquad \square$

## B    APPENDIX: PLOTS.

The graph representations of original view , isolated view of Bayes teacher and the student are shown in Fig. 3.

## C    APPENDIX: ALGORITHMS.

For presenting the algorithms, we slightly change the notation. We let $\boldsymbol{\alpha}(\mathbf{x}) = \boldsymbol{\alpha}(\mathbf{x}, \phi)$, $\mathbf{h}(\mathbf{x}) = \mathbf{h}(\mathbf{x}, \phi_1)$, $g(\mathbf{x}) = g(\mathbf{x}, \phi_2)$, where $\phi = \{\phi_1, \phi_2\}$, $\phi_1$ and $\phi_2$ are the parameters of prediction model $\mathbf{h}$ and concentration model $g$, respectively.

**KL.** With the training objective

$$
\min_{\phi} \quad -\mathbb{E}_{p_{\mathcal{D}'}(\mathbf{x})} \frac{1}{S} \sum_s \ln q(\mathcal{T}(\mathbf{x}; \boldsymbol{\theta}_s) | \mathbf{x}, \boldsymbol{\alpha}), \tag{9}
$$

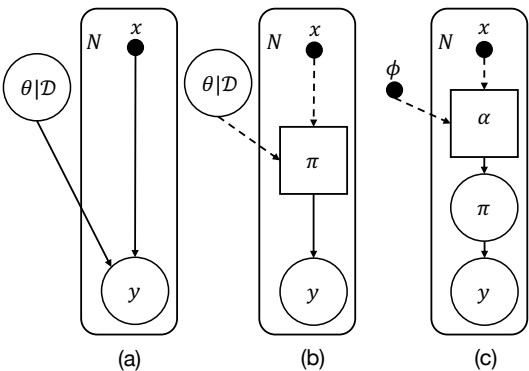

Figure 3: Graphical representation of probabilistic structure of the Bayes teacher (left and middle) and the student (right). Dashed edges denote deterministic dependence, box nodes are deterministic and circle nodes are stochastic. The left and middle graphs correspond to the LHS and RHS of (2), respectively.

The student model is updated by doing $\phi_1^{t+1} := \phi_1^t - \gamma^t(-\frac{1}{S}\sum_{\boldsymbol{\theta}_s} \nabla_{\phi_1} \ln q(\mathcal{T}(\mathbf{x}^*; \boldsymbol{\theta}_s)|\alpha(\mathbf{x}^*; \phi_1)))$ and $\phi_2^{t+1} := \phi_2^t - \gamma^t(-\frac{1}{S}\sum_{\boldsymbol{\theta}_s} \nabla_{\phi_2} \ln q(\mathcal{T}(\mathbf{x}^*; \boldsymbol{\theta}_s)|\alpha(\mathbf{x}^*; \phi_2)))$ alternately, where $\gamma^t$ is the learning rate at iteration $t$ and $\mathbf{x}^*$ is the input at this iteration.

**EMD.** With the following training objective,

$$\min_{\phi} \max_{w} \quad \mathbb{E}_{p_{\mathcal{D}'}(\mathbf{x})} \left[ \mathbb{E}_{p(\boldsymbol{\theta}|\mathcal{D})}[\psi(\mathcal{T}_{\mathbf{x}}(\boldsymbol{\theta}); w, \mathbf{x})] - \mathbb{E}_{q_{\mathbf{x}}}[\psi(\boldsymbol{\pi}; w, \mathbf{x})] \right] + \lambda \mathcal{R}(w),$$

---

**Algorithm 1:** OPU Training Algorithm with EMD

---

**Input**: Posterior samples: $\{\boldsymbol{\theta}_s\}_{s=1}^S$; OPU training data $\mathcal{D}'$; Gradient penalty coefficient $\lambda$; Number of training iterations: $T_{\text{stu}}$ and $T_{\text{wit}}$.

**while** $\phi$ *not converge* **do**
    Sample $\mathbf{x}^{(i)} \sim p_{\mathcal{D}'}(\mathbf{x})$
    /* Update Approximation                                              */
    **for** *iter in* $1 \dots T_{\text{stu}}$ **do**
        Sample $\{\boldsymbol{\pi}_{s'}\}_{s'=1}^{S'} \sim q(\boldsymbol{\pi}|\mathbf{x}^{(i)}, \phi)$
        $\mathcal{L}^{(i)}(\phi_1) = -\sum_{s'=1}^{S'} \psi(\boldsymbol{\pi}_{s'}; w, \mathbf{x}^{(i)})$
        $\phi_1 \leftarrow \text{Adam}(\nabla_{\phi_1}\mathcal{L}^{(i)})$
        Sample $\{\boldsymbol{\pi}_{s'}\}_{s'=1}^{S'} \sim q(\boldsymbol{\pi}|\mathbf{x}^{(i)}, \phi)$
        $\mathcal{L}^{(i)}(\phi_2) = -\sum_{s'=1}^{S'} \psi(\boldsymbol{\pi}_{s'}; w, \mathbf{x}^{(i)})$
        $\phi_2 \leftarrow \text{Adam}(\nabla_{\phi_2}\mathcal{L}^{(i)})$
    **end**
    /* Update Critic                                                      */
    **for** *iter in* $1 \dots T_{\text{wit}}$ **do**
        Sample $\{\boldsymbol{\pi}_{s'=1}^{S'}\} \sim q(\boldsymbol{\pi}|\mathbf{x}^{(i)}, \phi)$
        Compute $\mathcal{R}(v)$
        $\mathcal{L}^{(i)}(v) = \sum_{s=1}^{S} \psi(\mathcal{T}(\mathbf{x}; \boldsymbol{\theta}_s); w, \mathbf{x}^{(i)}) - \sum_{s'=1}^{S'} \psi(\boldsymbol{\pi}_{s'}; w, \mathbf{x}^{(i)}) + \lambda \mathcal{R}(w)$
        $v \leftarrow \text{Adam}(\nabla_v \mathcal{L}^{(i)})$
    **end**
**end**

---

**MMD.** The kernel mean embedding of $p_{\mathbf{x}}$ and $q_{\mathbf{x}}$ are given by $\mu_p = \mathbb{E}_{p_{\mathbf{x}}}[k(\boldsymbol{\pi}, \cdot)]$ and $\mu_q = \mathbb{E}_{q_{\mathbf{x}}}[k(\boldsymbol{\pi}, \cdot)]$. The training objective is then,

$$\min_{\phi} \|\mu_p - \mu_q\|_{\mathcal{H}_k}, \tag{10}$$

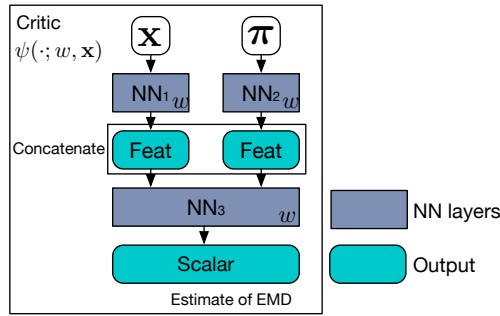

Figure 4: A brief example of the critic in EMD.

---

**Algorithm 2:** OPU Training Algorithm with MMD

---

**Input**: Posterior samples: $\{\boldsymbol{\theta}_s\}_{s=1}^S$; OPU training data $\mathcal{D}'$; Gradient penalty coefficient $\lambda$.
**while** $\phi$ *not converge* **do**

> Sample $\mathbf{x}^{(i)} \sim p_{\mathcal{D}'}(\mathbf{x})$
> Sample $\{\boldsymbol{\pi}_{q,s'}\}_{s'=1}^{S'} \sim q(\boldsymbol{\pi}|\mathbf{x}^{(i)}, \phi)$, get $\{\boldsymbol{\pi}_{p,s'}\}_{s'=1}^{S'} = \{\mathcal{T}(\mathbf{x}|\boldsymbol{\theta}_{s'})\}_{s'=1}^{S'}$
> $\mathcal{L}^{(i)}(\boldsymbol{\phi_1}) = \frac{1}{S'(S'-1)} \sum_{m\neq n}^{S'} k(\boldsymbol{\pi}_{q,m}, \boldsymbol{\pi}_{q,m}) + \frac{1}{S'(S'-1)} \sum_{m\neq n}^{S'} k(\boldsymbol{\pi}_{p,m}, \boldsymbol{\pi}_{p,m}) - \frac{2}{S'(S'-1)} \sum_{m,n=1}^{S',S'} k(\boldsymbol{\pi}_{q,m}, \boldsymbol{\pi}_{p,n})$
> $\phi_1 \leftarrow \mathrm{Adam}(\nabla_{\phi_1}\mathcal{L}^{(i)})$
> Sample $\{\boldsymbol{\pi}_{s'}\}_{s'=1}^{S'} \sim q(\boldsymbol{\pi}|\mathbf{x}^{(i)}, \phi)$
> $\mathcal{L}^{(i)}(\boldsymbol{\phi_1}) = \frac{1}{S'(S'-1)} \sum_{m\neq n}^{S'} k(\boldsymbol{\pi}_{q,m}, \boldsymbol{\pi}_{q,m}) + \frac{1}{S'(S'-1)} \sum_{m\neq n}^{S'} k(\boldsymbol{\pi}_{p,m}, \boldsymbol{\pi}_{p,m}) - \frac{2}{S'(S'-1)} \sum_{m,n=1}^{S',S'} k(\boldsymbol{\pi}_{q,m}, \boldsymbol{\pi}_{p,n})$
> $\phi \leftarrow \mathrm{Adam}(\nabla_{\phi_2}\mathcal{L}^{(i)})$

**end**

---

**Reparameterization:** To obtain efficient gradient estimator and reduce variance, we reparameterize the Dirichlet by an equivalent product of $K$ independent Gamma distributions. If $\tilde{\boldsymbol{\pi}} \sim \mathrm{PG}(\tilde{\boldsymbol{\pi}}|\boldsymbol{\alpha}) = \prod_{k=1}^K \mathrm{Gam}(\tilde{\pi}_k|\alpha_k)$, then $\boldsymbol{\pi} = (\sum_k \tilde{\pi}_k)^{-1} \tilde{\boldsymbol{\pi}} \sim \mathrm{Dir}(\boldsymbol{\pi}|\boldsymbol{\alpha})$. By Thm. 3 in (Arjovsky et al., 2017), in each $\mathcal{L}(\phi|\mathbf{x})$, the supremum is attained at $\psi_\mathbf{x}^* \in \mathrm{L}_1$ and the gradient is $\nabla_\phi \mathcal{L}(\phi|\mathbf{x}) = -\nabla_\phi \mathbb{E}_{q(\boldsymbol{\pi}|\mathbf{x},\phi)}[\psi_\mathbf{x}^*(\boldsymbol{\pi})]$. Then as noted by (Figurnov et al., 2018), the gradient $\nabla_\phi \mathcal{L}(\phi|\mathbf{x})$ can be implicitly computed without knowing the inverse of standardization function (e.g., CDF). Specifically, by Eq. 5 in (Figurnov et al., 2018), $\nabla_\phi \mathbb{E}_{q(\boldsymbol{\pi}|\mathbf{x};\phi)}[\psi_\mathbf{x}^*(\boldsymbol{\pi})] = \mathbb{E}_{\mathrm{PG}(\tilde{\boldsymbol{\pi}}|\mathbf{x};\phi)}[\nabla_{\tilde{\boldsymbol{\pi}}}\psi_\mathbf{x}^*(\boldsymbol{\pi})\nabla_\phi \tilde{\boldsymbol{\pi}}]$, where the first term $\nabla_{\tilde{\boldsymbol{\pi}}}\psi_\mathbf{x}^*(\boldsymbol{\pi})$ is computed via the chain rule and the second term $\nabla_\phi \tilde{\boldsymbol{\pi}}$ is obtained by solving a local diagonal linear system. Refer to (Figurnov et al., 2018) and references therein for details.

## D  APPENDIX: EXPERIMENTS

For the critic in EMD used in MNIST experiments, NN1 is a 784-256 MLP, NN2 is a 10-256 MLP and NN3 is a 512-256-1 MLP (see Fig. 4). For the critic in EMD used in MNIST experiments, NN1 is the same as convolutional layers used in Ba, NN2 is a 10-256 MLP and NN3 is a 512-256-1 MLP (see Fig. 4).

### D.1  BAYESIAN LOGISTIC REGRESSION

We test two models in this experiment: Polya Gamma (PG), CompactApprox approximating PG (CA-PG) and OPU approximating PG (OPU-PG). We draw 500 posterior samples from PG and train OPU with the following hyperparameters: number of epochs 100, learning rate for student. CA-PG is trained by first drawing 5000 samples from PG then evaluating the model with 50 randomly selected samples from them (same setup as CompactApprox). The random selection is repeated for $10^5$ times

Table 3: Results on Bayesian logistic regression models.

| Data | Model | MisC detection | | OOD detection | | Acc. | Time |
|------|-------|------|------|------|------|------|------|
| | | AUROC | AUPR | AUROC | AUPR | (%) | (sec.) |
| Pima | PG | **60.0** (Ent) | 24.2 (Ent) | 87.0 (Ent) | 76.7 (Ent) | **64.4** | $1.01 \pm 0.001$ |
| | CA-PG | 58.2 (Ent) | 24.2 (Ent) | 80.1 (Ent) | 74.5 (Ent) | 62.3 | $0.18 \pm 0.001$ |
| | OPU-PG | 59.7 (Ent) | **25.6** (Ent) | **100.0** (CM) | **100.0** (CM) | **64.4** | **0.01**$\pm 0.002$ |
| Spam | PG | **83.9** (Ent) | **24.3** (Ent) | 54.6 (Ent) | 53.5 (Ent) | **92.4** | $5.94 \pm 0.001$ |
| | CA-PG | 64.1 (Ent) | 24.2 (Ent) | 71.5 (Ent) | 67.5 (Ent) | 85.4 | $0.36 \pm 0.001$ |
| | OPU-PG | **83.9** (Ent) | 23.8 (Ent) | **99.7** (CM) | 99.3 (CM) | **92.4** | **0.01**$\pm 0.002$ |

Table 4: Results on Gaussian process classification models.

| Data | Model | MisC detection | | OOD detection | | Acc. | Time |
|------|-------|------|------|------|------|------|------|
| | | AUROC | AUPR | AUROC | AUPR | (%) | (second) |
| Pima | SGPMC | 65.3 (E) | **46.4** (E) | 96.7 (E) | 94.9 (E) | **79.3** | **0.003** |
| | SVGP | 64.3 (E) | 43.2 (E) | 96.0 (E) | 91.1 (E) | 77.1 | 0.004 |
| | OPU-SGPMC | **65.7** (E) | 44.4 (E) | **100.0** (C) | **100.0** (C) | 79.2 | 0.010 |
| Spam | SGPMC | **86.7** (E) | 37.8 (E) | 98.6 (E) | 97.6 (E) | **92.4** | 0.056 |
| | SVGP | 86.2 (E) | 33.3 (E) | 99.2 (E) | 98.5 (E) | 92.1 | 0.032 |
| | OPU-SGPMC | 86.5 (E) | **39.5** (E) | **100.0** (C) | **100.0** (C) | 92.0 | **0.011** |

and we pick the best group of samples. As the results for the three metrics are similar, we only show the results trained with KL divergence.

The results are shown in Table 3. OPU-PG maintains similar performance with the original PG on prediction accuracy and MisC detection. Meanwhile, OPU outperforms CA-PG on prediction accuracy, MisC and OOD detection. For OPU, CM outperforms other uncertainty measures at OOD detection, which indicates it captures the distributional uncertainty well. OPU performs better than the PG Bayes teacher. The reason might be that a parametric model is learned to approximate the ensemble of discrete samples, which could produce a smoother output distribution (regularization), leading to better performance. OPU also achieves a $\sim$100-600x speedup from the original PG.

## D.2 GAUSSIAN PROCESS

For GP, we use SGPMC (Hensman et al., 2015a) as the teacher, and SVGP (Hensman et al., 2015b) for comparison.

This experiment uses Pima and Spambase datasets as $\mathcal{D}^{\text{in}}$. Pima is a medical dataset with 769 data points and 9 dimensions. Spambase is a text dataset with 4601 data points and 57 dimensions for identifying spam email. We generate the same number of data points from a zero-mean multivariate Gaussian distribution for $\mathcal{D}^{\text{ood}}$. For each dataset, 10% of data points are uniformly selected into the testing set $\mathcal{D}^{\text{te}}$. We normalize the data by features with L2 norm. where $K$ is the number of classes. There are 3 models tested: SGPMC, OPU approximating SGPMC (OPU-SGPMC) and SVGP. SGPMC and SVGP are trained with $\frac{1}{10}|\mathcal{D}^{\text{in}}|$ data points randomly selected from $\mathcal{D}^{in}$ as inducing points. Then 500 samples over functions of $\mathcal{D}^{\text{in}}$ are generated from SGPMC. As the results for the three metrics are similar, we only show the results trained with KL divergence.

The results are presented in Table 4. On MisC detection and prediction accuracy, OPU has similar performance to SGPMC, which indicates the effectiveness of approximating prediction results with ensemble of samples in the nonparametric family. With the same number of inducing points, SVGP performs slightly worse than SGPMC, because it incurs a two-fold approximation. Measuring uncertainty with CM in OPU outperforms other measures and models, which indicates the sharpness of the logistic-normal distribution can be captured via the CM the designed Dirichlet.

SGPMC and SVGP are faster than OPU on Pima, but are slower than OPU on the larger Spambase. This is due to the static latency for setting up the GPU for OPU, which becomes the main time cost when the dataset is small (as in Pima). For the two GP methods, the computation time depends on the number of inducing points and the number of dimensions. Therefore, as the dataset becomes larger (Spambase), the computation time increases.

Table 5: Results on error/gap on EMNIST.

| Avg.Approx.Err. | Avg.Model.Err. | Avg.Amt.Gap | Avg.Amt.Err. |
|---|---|---|---|
| 0.0650 | 0.0601 | 0.0049 | 0.0053 |

### D.3 MORE EXPERIMENTS ON BAYESIAN NEURAL NETWORK

In this section, we present more experimental results/details on Bayesian NN.

**Computation of AUROC/AUPR:** First, we show how AUROC/AUPR is calculated. We use entropy (E) and max probability (P) of the particle mean, and differential entropy (D) of locally fitted student distribution as the measures for the teacher. We use entropy (E) and max probability (P) of the particle mean, and the scalar output of student concentration model (C) as the measures for the OPU student. For OOD detection, say $q_{\mathbf{x}_1}$ has higher E (or lower P or lower C) than $q_{\mathbf{x}_2}$, then $q_{\mathbf{x}_1}$ is more likely to be an OOD data. The value of E, P and C (D for the teacher) are computed for all testing data points (both in domain data and out-of-domain data), which rank the data points based on the numerical magnitude. The ROC curve is plotted by setting threshold on each magnitude and compute the True Positive Rate and False Positive Rate at each threshold. The PR curve is plotted similar by computing the Precision and Recall. Then the area under two curves (AUROC and AUPR) can be obtained. For misclassification detection, the similar calculation process is applied.

**Error/gap:** Second, we use the four types of error/gap under MMD defined in Sec. 2.4 to show the effectiveness of amortized approximation and the suitableness of using Dirichlet family. Specifically, the following measures are used:

- *Averaged total approximation error* (Avg.Approx.Err.): The averaged MMD between teacher's particles (for each x) and the predicted Dirichlet by OPU, i.e., $\frac{1}{N}\sum_{i=1}^{N}\mathrm{MMD}(q_{\mathbf{x}_i}, p_{\mathbf{x}_i})$.
- *Averaged total approximation error* (Avg.Model.Err.): The averaged MMD between teacher's particles (for each x) and locally fitted Dirichlet, i.e., $\frac{1}{N}\sum_{i=1}^{N}\mathrm{MMD}(p_{\mathbf{x}_i}, \bar{q}^*_{\mathbf{x}_i})$.
- *Averaged local amortization gap* (Avg.Amt.Gap): The difference between Avg.Approx.Err and Avg.Model.Err., i.e., $\Delta(\mathbf{x})$, defined in Eq. 7.
- *Averaged local amortization error* (Avg.Amt.Err.): We define this type of error to be the averaged MMD between locally fitted Dirichlet (for each x) and the predicted Dirichlet by OPU, i.e., $\frac{1}{N}\sum_{i=1}^{N}\mathrm{MMD}(q_{\mathbf{x}_i}, \bar{q}^*_{\mathbf{x}_i})$. The relation between Avg.Amt.Gap and Avg.Amt.Err. is given by Eq. 8.

We show the numerical results on EMNIST dataset by Table 5. It can be observed that the Avg.Amt.Err. 0.0053 is low and it bounds the Avg.Amt.Gap $\Delta(\mathbf{x})$=Avg.Approx.Err. − Avg.Model.Err=0.0049, which is consistent with Eq. 8. The approximation error is mainly determined by the model error, which turns out to be acceptably small. This is consistent with the analysis and also shows the effectiveness and suitableness of using Dirichlet family.

**Cifar10 results:** Third, we show the experimental results of OPU approximating Bayesian NN on Cifar10 dataset in Table 6. The NN model for teacher and student is standard VGG19 with the output dimension of concentration model to be 1. The MCDP teacher model is trained with SGD optimizer and a cyclical learning rate policy. The base learning rate is initialized to be 0.01 and linearly increased to 10x of learning rate, then decreases back to the base learning rate. The base learning rate is then scaled by half. The number of MCDP particles is 700. The OPU students are trained with KL/EMD/MMD using the corresponding algorithm for 200 epochs.

As the task is harder than MNIST, particles tends to be more spread over the corners. Under such case, it can be observed that the performance of EMD and MMD is much better than that of KL.

**Teachers with fewer samples:** Fourth, the performance of original MCDP/SGLD with fewer samples is shown in Table 7. We note that using few particles would affect the performance of Bayesian classifier, especially the performance on OOD detection. In real-world cases like automatic driving car where the robustness is critical, it is not worth to trade safety for speed. Therefore, OPU solves this issue by providing accurate estimation of predictive uncertainty with short evaluation time. Besides the speedup, OPU approximation provides a full distribution to characterize the predictive

Table 6: Results on BNN - Cifar10.

| Model | MisC detection | | OOD detection LSUN | | Acc. |
|---|---|---|---|---|---|
| | AUROC | AUPR | AUROC | AUPR | (%) |
| MCDP-KL | 92.2 (P) | 47.0 (P) | 90.5 (E) | 88.7 (E) | 92.4 |
| MCDP-EMD | 92.2 (P) | 47.0 (P) | 91.4 (D) | 89.1 (D) | 92.4 |
| MCDP-MMD | 92.2 (P) | 47.0 (P) | 91.0 (D) | 89.3 (D) | 92.4 |
| OPU-MCDP-KL | 87.2 (P) | 45.9 (P) | 86.1 (E) | 85.5 (E) | 89.9 |
| OPU-MCDP-EMD | 91.8 (E) | 46.9 (P) | 93.5 (C) | 92.0 (C) | 91.8 |
| OPU-MCDP-MMD | 91.3 (E) | 46.6 (P) | 92.9 (C) | 91.7 (C) | 91.8 |

Table 7: Results on MNIST - MCDP/SGLD with fewer samples.

| Model | MisC detection AUROC | Omniglot AUROC | SEMEION AUROC | Acc. (%) | Time s |
|---|---|---|---|---|---|
| MCDP | 96.1 (-1.2) | 98.5 (-0.9) | 82.7 (-4.1) | 94.9 (-3.0) | 132.1 |
| SGLD | 97.1 (-0.9) | 91.2 (-8.0) | 82.5 (-6.8) | 98.0 (-0.4) | 141.9 |

distribution, which is not available with particle approximation. This allows for better uncertainty measures such as differential entropy.

**Un-distilled teachers:** Finally, we show the detailed un-distilled MCDP/SGLD, whose elements might be covered by differential entropy of locally fitted Dirichlet in Table 1. The detailed results are shown in Table 8

# E  APPENDIX: SAMPLING

In this section, we illustrate the details for extracting samples from Bayesian logistic regression, Bayesian neural network and Gaussian process. Under some contexts, we use $\mathbf{X}$ and $\mathbf{Y}$ to collectively denote the inputs and outputs respectively for previous $\mathcal{D}$.

## E.1  BAYESIAN LOGISTIC REGRESSION

The Polya-Gamma (PG) scheme (Polson et al., 2013) is a data augmentation strategy that allows for a closed-form Gibbs sampler. In binary classification, i.e., $y_n \in \mathcal{Y} = \{1, 0\}$, let $\boldsymbol{\theta}$ be the regression coefficients with a Gaussian conditional conjugate prior $p(\boldsymbol{\theta}) \sim \mathcal{N}(\mathbf{0}, \boldsymbol{\Lambda}^{-1})$. The PG Gibbs sampler is composed of the following two conditionals,

$$\omega_n | \boldsymbol{\theta}, \mathbf{x}_n \quad \sim \quad \text{PG}(1, \mathbf{x}_n^{\mathsf{T}} \boldsymbol{\theta}) \tag{11}$$
$$\boldsymbol{\theta} | \boldsymbol{\omega}, \mathcal{D} \quad \sim \quad \mathcal{N}(\mathbf{m}_\omega, \mathbf{V}_\omega), \tag{12}$$

where $\omega_n$ is the augmenting data corresponding to the $n$th data point. The posterior conditional variance and mean are given by $\mathbf{V}_\omega = (\mathbf{X}^{\mathsf{T}} \Omega \mathbf{X} + \boldsymbol{\Lambda}^{-1})^{-1}$ and $\mathbf{m}_\omega = \mathbf{V}_\omega^{-1}(\mathbf{X}^{\mathsf{T}}(\mathbf{y} - \frac{1}{2}))$, respectively.

Given this formulation we will be able to collect samples from the posterior after a burn-in period. The posterior samples $\{\boldsymbol{\theta}\}_{s=1}^S$, together with dataset $\mathcal{D}'$, are used to train our approximation with goal defined in Eq. 10.

As an MCMC method, the PG augmentation scheme offers accurate samples. Other alternative methods such as local variational approximation can be employed, which are much faster but sacrifice accuracy.

## E.2  MONTE CARLO DROPOUT

A neural network with dropout applied before every weight layer was shown to be an approximation to probabilistic deep Gaussian process (GP) (Gal & Ghahramani, 2016). Let $q(\boldsymbol{\theta})$ to be the approximate distribution to the GP posterior. Here, $\boldsymbol{\theta} = \{\boldsymbol{\Theta}_i\}_{i=1}^L$ and $\boldsymbol{\Theta}_i$ is a parameter matrix of dimensions $K_i \times K_{i=1}$ for NN layer $i$. In this approximation, $q(\boldsymbol{\theta})$ can be defined through direct modification:

$$\boldsymbol{\Theta}_i = \mathbf{M}_i \text{diag}([z_{i,j}]_{i=1}^{K_i}), \tag{13}$$

Table 8: Results on MNIST - undistilled MCDP/SGLD.

| Model | MisC detection | | Omniglot | | SEMEION | | Acc. |
|-------|------|------|------|------|------|------|------|
| | AUROC | AUPR | AUROC | AUPR | AUROC | AUPR | (%) |
| MCDP | 97.3 (E) | 43.0 (E) | 99.2 (E) | 98.8 (P) | 86.8 (E) | 53.8 (P) | 97.9 |
| SGLD | 97.9 (E) | 46.2 (E) | 99.2 (E) | 99.6 (E) | 89.3 (E) | 46.8 (E) | 98.4 |

where $z_{i,j} \sim \text{Bern}(p_i)$ for $i \in [L]$ and $j \in [K_{i-1}]$, given some prior dropout probabilities $p_i$ and matrices $\boldsymbol{\Theta}_i$ are treated as variational parameters. The binary variable $z_{i,j} = 0$ indicates that unit $j$ in layer $i-1$ being dropped out as an input to layer $i$. The predictive mean of this approximation is given by $\mathbb{E}[\mathbf{y}|\mathbf{x}] \approx \frac{1}{S} \sum_{s=1}^{S} f(\mathbf{x}, \hat{z}_{1,s}, \ldots, \hat{z}_{L,s})$, which hence referred to as MC dropout (MCDP).

We use OPU to approximate the uncertainty induced by $q(\boldsymbol{\theta})$. A sample in the MC ensemble is given by $\boldsymbol{\theta}_s = \{\boldsymbol{\Theta}_{i,s}\}_{i=1}^{L} = \mathbf{M}_i \text{diag}([z_{i,j}]_{i=1}^{K_i})$. We set $\boldsymbol{\phi}_1 = \{\boldsymbol{\Phi}_i\}_{i=1}^{L} Z$ to the mean of $\boldsymbol{\theta}_s$, i.e., $\boldsymbol{\Phi}_i = \frac{1}{S} \sum_{s=1}^{S} \boldsymbol{\Theta}_{i,s} = \mathbf{M}_i \frac{1}{S} \sum_{s=1}^{S} \text{diag}([z_{i,j,s}]_{i=1}^{K_i})$.

MCDP provides a simple way of approximating Bayesian inference through dropout sampling. However, it still introduces a variational approximation to exact Bayesian posterior. Therefore, we further include a more accurate way to generate MC ensemble - stochastic gradient Langevin dynamics (SGLD).

### E.3 VANILLA SGLD

SGLD enables mini-batch MC sampling from the posterior via adding a noise step to SGD (Welling & Teh, 2011). We choose the "vanilla" version of SGLD in our approximation. Specifically, we start training of $f_{\boldsymbol{\theta}}(\cdot)$ from $\boldsymbol{\theta}^{(0)}$. In each epoch with mini-batch size $B$,

$$\boldsymbol{\theta}^{(t+1)} = \boldsymbol{\theta}^{(t)} + \epsilon_t \nabla \log p(\boldsymbol{\theta}^{(t)}|\mathcal{D}) + \eta_t \tag{14}$$

$$= \boldsymbol{\theta}^{(t)} + \epsilon_t \nabla (\log p(\boldsymbol{\theta}^{(t)}) + \sum_{b=1}^{B} \log p(\mathbf{y}_b|\mathbf{x}_b, \boldsymbol{\theta}^{(t)})) + \eta_t, \tag{15}$$

where $B$ is the size of a mini-batch and $\eta_t \sim \mathcal{N}(0, 2\epsilon_t \mathbf{I})$. After SGLD converges at step $T$, the samples $\boldsymbol{\theta}_s = \boldsymbol{\theta}^{(T+s)}$ is collected by running the training process for another $S$ iterations. We use averaged samples as parameter of $h$, i.e., $\phi_1 = \frac{1}{S} \sum_{s=1}^{S} \boldsymbol{\theta}_s$ and train OPU by Eq. 10 with this ensemble.

### E.4 MONTE CARLO GAUSSIAN PROCESS

We apply the OPU framework to the GP framework, which demonstrate its use on a non-parametric classifier. Let data $\mathcal{D}$ be split into as input matrix $\mathbf{X}$ and output matrix $\mathbf{Y}$. We consider GP prior over the space of functions, i.e., $\boldsymbol{\mu} \sim \mathcal{GP}(0, \mathcal{K})$. where $\mathcal{K}$ is a positive definite kernel controlling the prior belief on smoothness. Existing techniques allow us to compute $q(\boldsymbol{\mu})$ which approximates $p(\boldsymbol{\mu}|\mathcal{D})$ and is comparable to previous $q(\boldsymbol{\theta})$ under a parametric model. In a classification task, the posterior $p(\boldsymbol{\mu}|\mathcal{D})$ can be sampled via MCMC (Hensman et al., 2015a) or approximated, e.g., via variational approximation (Hensman et al., 2015b). Let $\boldsymbol{\mu}^*$ be a shorthand for $\boldsymbol{\mu}_{\mathbf{x}}$ and $\boldsymbol{\pi}$ is then defined as $\mathcal{S}(\boldsymbol{\mu}^*)$. If Gaussian variational approximation is used, the marginal posterior $p(\boldsymbol{\mu}^*|\mathbf{x}, \mathcal{D})$ at $\mathbf{x}$ induces a logistic-normal distribution for $p(\boldsymbol{\pi}|\mathbf{x}, \mathcal{D})$. In our approximation, we obtain samples $\{\boldsymbol{\mu}_s^*\}_{s=1}^{S}$ from $p(\boldsymbol{\mu}^*|\mathbf{x}, \mathcal{D})$. The optimization goal is the same as that in the NN case with a different target distribution defined as

$$\hat{p}(\boldsymbol{\pi}|\mathbf{x}, \mathcal{D}) = \frac{1}{S} \sum_{s=1}^{S} \delta(\boldsymbol{\pi} - \mathcal{S}(\boldsymbol{\mu}_s^*)). \tag{16}$$

