# OpenReview forum: "Accelerating Monte Carlo Bayesian Inference via Approximating Predictive Uncertainty over the Simplex"
_ICLR.cc/2020/Conference — Reject_

### Official Review · AnonReviewer4 · 2019-10-20
**Official Blind Review #4**

**Rating:** 3

**Review:**

Overall I liked several results presented in this paper. The findings in Figure 2 gives clear illustration on how Bayesian classification models distinguish between in-distribution difficult-to-classify data and out-of-distribution data, namely uncertain predicted mean and large predicted variance. Though I believe this eventually depends on what kind of "kernel"s are used to correlate data points in the prior, throughout the paper I assume meaningful "kernel"s are used (for Bayesian NNs this is rooted in the inductive bias of neural networks).

Another result that I liked is in experiments we can clearly see the advantage of considering the bayesian predictive distribution over a single predictive mean. As demonstrated by BDK-SGLD vs. BDK-DIR-SGLD.

The proposed idea is a simple and meaningful improvement over previous works. Though the contribution is quite limited, the authors present it with great clarity, which I appreciated. However, I do find some discussions in the paper unnecessary and would expect for more technical contributions. For example, I didn't see the argument for the whole section discussing amortization gap. Everything seems straightforward given the hypothesis F is of enough capacity, which obviously does not hold in practice.

Many other concerns are summarized below:
* What if the teacher predictive distribution is far unlike a Dirichlet? How much is the discrepancy between teacher and student predictive distribution? Theoretical or empirical evidence is needed for this modeling choice.
* One importance advantage of Bayesian classification models is that they can capture the covariance between predictions  of different data points. By amortization this advantage no longer exists.
* In the paper the authors keep mentioning that the method can be applied to GPs but I don't see experiments or algorithms for it?
* The concentration model is parameterized using an exponential activation, how does this activation affect the performance?
* The distilling process is done on a held-out dataset. Which may not be wanted because an advantage of Bayesian classification models (eg. GPs) is that all hyperparameters can be automatically selected by marginal likelihoods and don't need a held-out validation set.
* MMD/wasserstein distances are cool but they require also samples from the student, which adds more variance to the distillation process.
* The experiment setup is extremely unclear to me. What is "uncertainty measures", are they used as metrics for detecting out-of-distribution data, how are AUROC/AUPR calculated using the uncertainty measures? I can guess the meaning but the paper should be more clear about this.
* I found most numbers convincing except that sometimes BDK-SGLD outperforms BDK-DIR-SGLD, if I understand it right, the predicted mean of BDK-DIR-SGLD should be as good as BDK-SGLD?

Minor:
* On page 4, above Eq. (4) there is a broken figure link.
* On Page 7, "To save space, we only present the best performing uncertainty measure (E, P or C)". What is "C" here?

**Experience Assessment:**

I have published one or two papers in this area.

**Review Assessment: Checking Correctness Of Derivations And Theory:**

I carefully checked the derivations and theory.

**Review Assessment: Checking Correctness Of Experiments:**

I assessed the sensibility of the experiments.

**Review Assessment: Thoroughness In Paper Reading:**

I read the paper at least twice and used my best judgement in assessing the paper.

---

> ### Author Response · Authors · 2019-11-09
> **Response to Reviewer #4, part 1.**
>
> Review #4
> We thank the reviewer for the comments and would like to answer the questions as follows:
>
> Q1: However, I do find some discussions in the paper unnecessary and would expect for more technical contributions. For example, I didn't see the argument for the whole section discussing amortization gap. Everything seems straightforward given the hypothesis F is of enough capacity, which obviously does not hold in practice.
>
> To understand the amortized approximation problem better, we show the total approximation error can be decomposed into model error and amortization gap, and that the amortization gap can be reduced to zero given enough capacity. The analysis aims to show that the idea of “amortization” is appropriate in our particular scenario. However, in our application, a strong model doesn’t always translate to small approximation error. For example, consider inference gap in VAE [1]. The local amortization gap is also useful as a general evaluation metric in the amortized knowledge distillation problems. The assumption of F having enough capacity is an assumption also used in the analyses of GAN and WGAN [2,3].
> In the experiment (Sec. 4.2 on page 7), we show that the amortized student model approximates each local distribution (that with only model error) in high-fidelity, indicating a low amortization loss. We add an experiment on the EMNIST dataset to verify the decomposition of total approximation error empirically.
>
> More results on EMNIST OPU-MCDP-MMD:
> Average approximation error [Eq. 7 on page 5]: $\frac{1}{N}\sum_{i=1}^{N} MMD(q_{\mathbf{x}_i}, p_{\mathbf{x}_i})$.  The averaged MMD between teacher’s particles (for each x) and the predicted Dirichlet by OPU: 6.5*10^(-2).
> Average model error [Eq. 7 on page 5]: $\frac{1}{N}\sum_{i=1}^{N} MMD(p_{\mathbf{x}_i}, \bar{q}_{\mathbf{x}_i}^\ast)$. The averaged MMD between teacher’s particles and locally fitted Dirichlet: 6.01*10^(-2).
> Average local amortization error: $\frac{1}{N}\sum_{i=1}^{N} MMD(q_{\mathbf{x}_i}, \bar{q}_{\mathbf{x}_i}^\ast)$. The averaged MMD between locally fitted Dirichlet (for each x) and the predicted Dirichlet by OPU: 5.3*10^(-3). Note that the “error” is different from “amortization gap” Δ(x) defined in the paper [Sec. 2.4 on page 5]. The relationship between “amortization error” defined here and “amortization gap” is given by Eq. 8 [Sec 2.4 on page 5].
>
> It can be observed that the local amortization error is low, even with a student model having limited capacity capacity. This effectively shows that the function space considered is enough to cover the essential target, which is in fact “of enough capacity”.
> The local amortization error 5.3*10^(-3) bounds the local amortization gap Δ(x) = Avg.Apx.Err – Avg.Mdl.Err = 4.9*10^(-3), which is consistent with Eq. 8 [Sec 2.4 on page 5].
>
> The approximation error is mainly determined by the model error, which turns out to be acceptably small. This is consistent with the analysis and also shows the effectiveness and suitableness of using Dirichlet family.
>
> Q2: What if the teacher predictive distribution is far unlike a Dirichlet? How much is the discrepancy between teacher and student predictive distribution? Theoretical or empirical evidence is needed for this modeling choice.
>
> Using a Dirichlet for the student is modeling choice, similar to assuming Gaussian posteriors for variational approximations. Our framework is general and any student distribution can be adopted, e.g., generalized Dirichlet, mixture of Dirichlets. To do this requires: 1) a suitable parametrization of the model that can capture uncertainty; 2) deriving/computing the approximation loss (KL, MMD, EMD); 3) reparameterization trick of expectations for efficient gradient estimation. The algorithms of KL, EMD and MMD as well as the analysis still apply.
>
> EMNIST is a more challenging dataset compared with Pima, Spambase, MNIST and Cifar10 as the number of classes is larger (47). The empirical evidence for the choice of Dirichlet is in this experiment, where we obtain nearly the same performance (accuracy) as using particles, while obtaining better OOD performance. We also show that the approximation error is low (see Q1), and thus the Dirichlet is a good fit for the teacher’s predictive distribution in this case.
>
> References:
> [1] Kingma, Diederik P., and Max Welling. "Auto-encoding variational bayes." arXiv preprint arXiv:1312.6114 (2013).
> [2] Goodfellow, Ian, et al. "Generative adversarial nets." Advances in neural information processing systems. 2014.
> [3] Arjovsky, Martin, Soumith Chintala, and Léon Bottou. "Wasserstein generative adversarial networks." International conference on machine learning. 2017.

---

> > ### Author Response · Authors · 2019-11-09
> > **Response to Reviewer #4, part 2.**
> >
> > Q3: One importance advantage of Bayesian classification models is that they can capture the covariance between predictions of different data points. By amortization this advantage no longer exists.
> >
> > This disadvantage is true but common for all distillation methods [1,2]. How to extend the distillation for a single input to more simultaneous inputs to model covariance in the predictions an interesting topic for future work.
> >
> > Q4: In the paper the authors keep mentioning that the method can be applied to GPs but I don't see experiments or algorithms for it?
> >
> > Due to space constraints we put the algorithms and experiments related to GPs in the appendix. Appendix E.4 shows how to get particles from G, and Appendix D.2 present the experimental results for GP. The algorithms are obtained by replacing samples with GP particles in Algorithm 1 and Algorithm 2.
> >
> > Q5: The concentration model is parameterized using an exponential activation, how does this activation affect the performance?
> >
> > We also tried softplus as the activation, and the performance was similar to using the exponential activation. Because the concentration value is non-negative, we require the output activation for the concentration model to be non-negative and monotonically increasing.
> >
> > Q6: The distilling process is done on a held-out dataset. Which may not be wanted because an advantage of Bayesian classification models (eg. GPs) is that all hyperparameters can be automatically selected by marginal likelihoods and don't need a held-out validation set.
> >
> > We refer to the "held-out" dataset D’ for training OPU so as to distinguish it from the training set of the teacher. Actually, in all the experiments, for fairness (BDK, CompactApprox and SVGP are not trained on held-out set), the dataset used to train OPU is the same as the teacher's training set. See the first sentence of “Data and Evaluation Metrics” on page 7. To clarify, the “held-out’’ dataset doesn’t mean validation set. The aim of this dataset is not to facilitate selection of hyperparameters, but to better capture uncertainties on points that do not appear in training data. To avoid confusion, we will rename the D’ dataset as “the OPU training dataset”.
> >
> > Q7: MMD/wasserstein distances are cool but they require also samples from the student, which adds more variance to the distillation process.
> >
> > This is true that samples from the students are required. We use the reparameterization technique in our framework to reduce the variance (see “Reparameterization” on page 5, and Appendix C for details).
> >
> > Q8: The experiment setup is extremely unclear to me. What is "uncertainty measures", are they used as metrics for detecting out-of-distribution data, how are AUROC/AUPR calculated using the uncertainty measures? I can guess the meaning but the paper should be more clear about this.
> >
> > We use entropy (E) and max probability (P) of the particle mean, and differential entropy (D) of locally fitted student distribution as the measures for the teacher. We use entropy (E) and max probability (P) of the particle mean, and the scalar output of student concentration model (C) as the measures for the OPU student. For OOD detection, say q_x1 has higher E (or lower P or lower C) than q_x2, then q_x1 is more likely to be an OOD data. The value of E, P and C (D for the teacher) are computed for all testing data points (both in domain data and out-of-domain data), which rank the data points based on the numerical magnitude. The ROC curve is plotted by setting threshold on each magnitude and compute the True Positive Rate and False Positive Rate at each threshold. The PR curve is plotted similar by computing the Precision and Recall. Then the area under two curves (AUROC and AUPR) can be obtained. For misclassification detection, the similar calculation process is applied.
> >
> > Q9: I found most numbers convincing except that sometimes BDK-SGLD outperforms BDK-DIR-SGLD, if I understand it right, the predicted mean of BDK-DIR-SGLD should be as good as BDK-SGLD?
> >
> > The reason might be that the parameterization of BDK-DIR without disentangling the mean and concentration is harder to learn. In comparison, BDK only approximates the mean of particles.
> >
> > Q10: On Page 7, "To save space, we only present the best performing uncertainty measure (E, P or C)". What is "C" here?
> >
> > C is the scalar output of the concentration model, which can be directly used as the uncertainty measure (See Para. 2, Page 4 and Para. 1, Page 7 ).
> >
> > We solved the minor issues and submitted a revision.
> >
> > [1] Bayesian dark knowledge. Advances in Neural Information Processing Systems 28, pp. 3438–3446.
> > [2] Distilling the knowledge in a neural network. In NIPS Deep Learning and Representation Learning Workshop, 2015.

---

### Official Review · AnonReviewer1 · 2019-10-21
**Official Blind Review #1**

**Rating:** 3

**Review:**

Thank the authors for your detailed rebuttal. I agree with the authors that the proposed method acts as a useful tool for "real-time evaluation of induced predictive uncertainty", and the experiments also validate that the method indeed achieves comparable performance with smaller computations. But for now, I am inclined to not change my score.

###################


Bayesian models maintain the posterior distribution for predictions, which might bring up big computational costs of multiple forwards or big memory costs of multiple particles. To resolve the computational and memory issues at predictions, this paper proposes to distill Bayesian models into an amortized prediction model, avoiding the original multiple forwards. Specifically, in classification, they distill the predictive probabilities into an amortized Dirichlet distribution. They evaluated different distillation metrics, including KL divergence, Earth moving distance, and Maximum mean discrepancy.  Empirically, they evaluate the proposed method over out-of-distribution detection. They demonstrate that their method achieves comparable performance with much speedup.

Strengths,
1, This paper is well-written and the ideas are well-presented. They evaluated the proposed method over different Bayesian models (MCDP & SGLD) as well different metrics (KL, EMD, MMD), and demonstrate the effectiveness of their method. Overall, this paper is very comprehensive.
2, As evaluated and validated in the experiments, the proposed method vastly reduces the inference time at test phase.

Weakness,
1, The paper kind of lacks of novelty. Basically the proposed method distills a Bayesian models into an amortized Dirichlet distribution, which is straightforward.
2, The baselines such as MCDP-KL, MCDP-EMD are strange, it is wired why you would distill the predictive distribution of a single point to a Dirichlet distribution. And I think it is probably unfair, as distilling the single-point distribution to the Dirichlet under KL, EMD, MMD might require large amount of particles, which they don't have.
3, Related to (2), more baselines should be compared with to better demonstrate the method's effectiveness. 1) performance of the un-distilled MCDP and SGLD models. 2) BDK and DPN for the MCDP models. 3) MCDP and SGLD with fewer particles. The paper claims to achieve 500x speed up, while I reckon the performance of MCDP and SGLD won't deteriorate a lot if you use only fewer particles.
4, It would be interesting to see experiments other than out-of-distribution detection, such as calibration.

Minor Issues,
1, The paper has several un-complied references, such as above eq(4) and appendix D.
2, The \Tau(x | \theta) in Figure 1 is a typo.
3, Assumption 1 should be put forward to the main articles for comprehensiveness of Lemma 1.

**Experience Assessment:**

I have published one or two papers in this area.

**Review Assessment: Checking Correctness Of Derivations And Theory:**

I carefully checked the derivations and theory.

**Review Assessment: Checking Correctness Of Experiments:**

I assessed the sensibility of the experiments.

**Review Assessment: Thoroughness In Paper Reading:**

I read the paper thoroughly.

---

> ### Author Response · Authors · 2019-11-09
> **Response to Reviewer #1, part 1.**
>
> We thank the reviewer for the comments and would like to answer the questions as follows:
>
> Q1: Lacks of novelty and straight forward.
>
> This paper proposes a framework that solves the practical problem of real-time evaluation of induced predictive uncertainty. Different from previous knowledge distillation [1,2], we provide a new view of induced distribution \pi which isolates the dependence between y and x, as shown by the graphic model in Fig. 3(b) in the Appendix. The “isolation view” is meaningful not only in classification, but also in all applications where predictive uncertainty are required, e.g., image object detection and segmentation. The “isolation view” also enables a richer characterization of student model (all previous works use a simple categorical distribution). As this kind of distillation is an unexplored problem, different evaluation metrics are considered and adapted to our framework. We also propose use the unamortized version to study how amortization affects the approximation, which decomposes the total approximation error into model error and amortization gap (Eq.7, Page 5). (see the new experimental results on EMNIST below) The local amortization gap can be used as an evaluation metric for amortized approximation. This also appears to be novel to the literature.
>
> The student distribution does not necessarily need to be a Dirichlet. The framework allows to use various choices of student distribution, and the algorithms based on KL, EMD and MMD as well as the analysis still apply.
>
> More results on EMNIST OPU-MCDP-MMD:
> Average approximation error [Eq. 7 on page 5]: $\frac{1}{N}\sum_{i=1}^{N} MMD(q_{\mathbf{x}_i}, p_{\mathbf{x}_i})$.  The averaged MMD between teacher’s particles (for each x) and the predicted Dirichlet by OPU: 6.5*10^(-2).
> Average model error [Eq. 7 on page 5]: $\frac{1}{N}\sum_{i=1}^{N} MMD(p_{\mathbf{x}_i}, \bar{q}_{\mathbf{x}_i}^\ast)$. The averaged MMD between teacher’s particles and locally fitted Dirichlet: 6.01*10^(-2).
> Average local amortization error: $\frac{1}{N}\sum_{i=1}^{N} MMD(q_{\mathbf{x}_i}, \bar{q}_{\mathbf{x}_i}^\ast)$. The averaged MMD between locally fitted Dirichlet (for each x) and the predicted Dirichlet by OPU: 5.3*10^(-3). Note that the “error” is different from “amortization gap” Δ(x) defined in the paper [Sec. 2.4 on page 5]. The relationship between “amortization error” defined here and “amortization gap” is given by Eq. 8 [Sec 2.4 on page 5].
>
> It can be observed that the local amortization error 5.3*10^(-3) is low and it bounds the local amortization gap Δ(x) = Avg.Apx.Err – Avg.Mdl.Err = 4.9*10^(-3), which is consistent with Eq. 8 [Sec 2.4 on page 5].
> The approximation error is mainly determined by the model error, which turns out to be acceptably small. This is consistent with the analysis and also shows the effectiveness and suitableness of using Dirichlet family.
>
> Q2: The "single-point" baselines are strange.
>
> We argue that this baseline is fair. Let the ‘single-point distribution’ be understood as the ‘local’ distribution for each input x we have defined, either local induced conditional distribution or local Dirichlet approximation.
> This baseline is consistent with the analysis, where the approximation loss equals the model loss and the amortization loss is zero, which should be the best performance OPU can achieve (within Dirichlet family) theoretically.
> When training OPU, we first extract 700 posterior samples from the pretrained MCDP/ SGLD model. (for fairness, the baselines and OPU use the same set of posterior samples) For each input x, this induces 700 particles over the simplex for OPU to approximate.
> When training each local distribution, for each x, a Dirichlet with a k-dim (k=10 for MNIST and Cifar10, k=47 for EMNIST) vector parameter is fitted on the 700 particles induced by the 700 posterior samples, which is enough particles to learn the Dirichlet well. The vector is also disentangled into a probability vector and a concentration scalar to be consistent, with no neural network parameterizing them (no amortization).
>
> References:
> [1] Bayesian dark knowledge. Advances in Neural Information Processing Systems 28, pp. 3438–3446.
> [2] Distilling the knowledge in a neural network. In NIPS Deep Learning and Representation Learning Workshop, 2015.

---

> > ### Author Response · Authors · 2019-11-09
> > **Response to Reviewer #1, part 2.**
> >
> > Q3: More experimental results: 1) un-distilled MCDP and SGLD models. 2) BDK and DPN for the MCDP models. 3) MCDP and SGLD with fewer particles.
> >
> > 1) The performance of un-distilled MCDP and SGLD model is already given in the experiment.
> > For a clear illustration, we show the performance of un-distilled MCDP/SGLD here.
> >
> > MCDP
> > MisC. AUROC 97.3 (E), AUPR 43.0 (E). OOD1. AUROC 99.2 (P) 98.8 (P) OOD2. AUROC 86.8 (E) 53.8 (P)
> >
> > SGLD
> > MisC. AUROC 97.9 (E), AUPR 46.2 (E). OOD1. AUROC 99.2 (E) 99.6 (E) OOD2. AUROC 89.3 (E) 46.8 (E)
> >
> > To avoid the model error, the numbers of MCDP are the same with MCDP-(KL/EMD/MMD) in terms of entropy (E) and maximum probability (P) of particle mean. The differential entropy (D) are from the fitted local distribution with KL/EMD/MMD where model error is involved. In table 1, we only show the best out of (E/P/D). Same for SGLD.
> >
> > 2) BDK results for MCDP.
> > MisC. Detection: AUROC 86.9 (E) AUPR 41.1 (E)
> > OOD omniglot: AUROC 47.5 (P) AUPR 44.1 (P)
> > OOD SEMEION: AUROC 43.3 (P) AUPR 47.2 (P)
> > As DPN is not designed to approximate a Bayes teacher, there is no DPN-MCDP and DPN-SGLD. The performance of DPN is already shown in Table 1.
> >
> > 3) MCDP and SGLD with fewer particles.
> > MCDP half samples:
> > Acc. 94.9 (-3.0%). MisC.AUROC: 96.1 (-1.2) OOD1.AUROC. 98.5 (-0.9) OOD2.AUROC. 82.7 (-4.1) Test time: 132.1 (s) (OPU has 298x speed up.)
> >
> > SGLD half samples:
> > Acc. 98.0 (-0.4%). MisC.AUROC: 97.1 (-0.9) OOD1.AUROC. 91.2 (-8.0) OOD2.AUROC. 82.5 (-6.8) Test time: 141.9 (s) (OPU has 320x speed up.)
> >
> > We note that using few particles would affect the performance of Bayesian classifier, especially the performance on OOD detection. In real-world cases like automatic driving car where the robustness is critical, it is not worth to trade safety for speed (see the fatality of assisted driving system [1]). Therefore, OPU solves this issue by providing accurate estimation of predictive uncertainty with short evaluation time. Besides the speedup, OPU approximation provides a full distribution to characterize the predictive distribution, which is not available with particle approximation. This allows for better uncertainty measures such as differential entropy.
> >
> > Q4: It would be interesting to see experiments other than out-of-distribution detection, such as calibration.
> >
> > We will add the calibration experiments.
> >
> > We solved the minor issues and submitted a revision.
> >
> > References:
> > [1] "What uncertainties do we need in bayesian deep learning for computer vision?." Advances in neural information processing systems. 2017.

---

### Official Review · AnonReviewer2 · 2019-10-26
**Official Blind Review #2**

**Rating:** 6

**Review:**

This paper studies the problem of avoiding Monte Carlo (MC) estimate for the predictive distribution during the test for Bayesian methods. MC estimate will incur multiple passes where the number of passes depends on the number of samples and therefore the cost can be huge. The authors propose One-Pass Uncertainty (OPU) methods to approximate the predictive distribution through distillation. Experiments on Bayesian neural networks are conducted to demonstrate the proposed method.

Quality:
The proposed method appears to be technically sound. The view of approximating the predictive distribution over simplex is interesting and may inspire future studies under this formulation. Although the restriction of the student distribution to be tractable seems to limit the design of the student model significantly. And this restrictive distribution family may cause large amortization error, as suggested by Lemma 1 in the paper.

The experiments are well-conducted, and the proposed method is well-evaluated.

Significance:
This paper studies an important problem in Bayesian machine learning and the proposed method can be combined with many Bayesian methods to reduce the computational cost during the test.

Originality:
As far as I know, the method is novel. The related work is adequately cited.

Clarity:
This paper is well-written and easy to follow.



**Experience Assessment:**

I do not know much about this area.

**Review Assessment: Checking Correctness Of Derivations And Theory:**

I did not assess the derivations or theory.

**Review Assessment: Checking Correctness Of Experiments:**

I assessed the sensibility of the experiments.

**Review Assessment: Thoroughness In Paper Reading:**

I read the paper at least twice and used my best judgement in assessing the paper.

---

> ### Author Response · Authors · 2019-11-09
> **Response to Reviewer #2**
>
> We thank the reviewer for the comments and would like to answer the questions as follows:
>
> Q1:Although the restriction of the student distribution to be tractable seems to limit the design of the student model significantly. And this restrictive distribution family may cause large amortization error, as suggested by Lemma 1 in the paper.
>
> The restrictive distribution family causes “large” total approximation error if the ground truth induced distribution is considerably different from a Dirichlet. However the amortization error is low and depends on capacity of neural network used to construct the approximation [point 2 in Para 1. Sec 2.2 on page 3]. We show the three types of errors with the experimental results on EMNIST.
>
> Results on EMNIST OPU-MCDP-MMD:
> Average approximation error [Eq. 7 on page 5]: $\frac{1}{N}\sum_{i=1}^{N} MMD(q_{\mathbf{x}_i}, p_{\mathbf{x}_i})$.  The averaged MMD between teacher’s particles (for each x) and the predicted Dirichlet by OPU: 6.5*10^(-2).
> Average model error [Eq. 7 on page 5]: $\frac{1}{N}\sum_{i=1}^{N} MMD(p_{\mathbf{x}_i}, \bar{q}_{\mathbf{x}_i}^\ast)$. The averaged MMD between teacher’s particles and locally fitted Dirichlet: 6.01*10^(-2).
> Average local amortization error: $\frac{1}{N}\sum_{i=1}^{N} MMD(q_{\mathbf{x}_i}, \bar{q}_{\mathbf{x}_i}^\ast)$. The averaged MMD between locally fitted Dirichlet (for each x) and the predicted Dirichlet by OPU: 5.3*10^(-3). Note that the “error” is different from “amortization gap” Δ(x) defined in the paper [Sec. 2.4 on page 5]. The relationship between “amortization error” defined here and “amortization gap” is given by Eq. 8 [Sec 2.4 on page 5].
>
> It can be observed that the local amortization error 5.3*10^(-3) is low and it bounds the local amortization gap Δ(x) = Avg.Approx.Err – Avg.Model.Err = 4.9*10^(-3), which is consistent with Eq. 8 [Sec 2.4 on page 5].
> The approximation error is mainly determined by the model error, which turns out to be acceptably small. This is consistent with the analysis and also shows the effectiveness and suitableness of using Dirichlet family.
>
> More experiments on MCDP-Cifar10. (Code available via the code link)
> Method                 || MisC. AUROC|| MisC. AUPR || OOD. AUROC || OOD. AUPR || Acc
> MCDP-KL:             ||     92.2 (P)      ||  47.0 (P)        ||   90.5 (E)         ||   88.7 (E)       || 92.4
> MCDP-EMD:         ||     92.2 (P)      ||  47.0 (P)        ||   91.4 (D)        ||   89.1 (D)       || 92.4
> MCDP-MMD:        ||     92.2 (P)      ||  47.0 (P)       ||   91.0 (D)        ||    89.3 (D)      || 92.4
> OPU-MCDP-KL:     ||     87.2 (P)     ||  45.9 (P)       ||   86.1 (E)         ||   85.5 (E)       || 89.9
> OPU-MCDP-EMD:||     91.8 (E)      ||  46.9 (P)       ||   93.5 (C)         ||   92.0 (C)       || 91.8
> OPU-MCDP-MMD:||    91.3 (E)      ||  46.6 (P)       ||   92.9 (C)         ||   91.7 (C)       || 91.8

---

### Author Response · Authors · 2019-11-15
**A new revision is submitted.**

We thank all reviewers for the constructive feedback. We revised the paper according to the feedbacks and submitted a revision. The major changes are highlighted in blue, either in main body or in appendix. Some new experimental results and analysis are shown in appendix D.3, page 14-15.

---

### Decision · Program_Chairs · 2019-12-19

**Decision:**

Reject

**Comment:**

This paper proposes to speed up Bayesian deep learning at test time by training a student network to approximate the BNN's output distribution. The idea is certainly a reasonable thing to try, and the writing is mostly good (though as some reviewers point out, certain sections might not be necessary). The idea is fairly obvious, though, so the question is whether the experimental results are impressive enough by themselves to justify acceptance. The method is able to get close to the performance achieved by Monte Carlo estimators with much lower cost, although there is a nontrivial drop in accuracy. This is probably worth paying if it achieves 500x computation reduction as claimed in the paper, though the practical gains are probably much smaller since Monte Carlo methods are rarely used with 500 samples. Overall, this seems a bit below the bar for ICLR.